# A Lower Bound of Hash Codes' Performance

**Xiaosu Zhu[1]**
xiaosu.zhu@outlook.com

**Jingkuan Song[1]***
jingkuan.song@gmail.com

**Yu Lei[1]**
leiyu6969@gmail.com

**Lianli Gao[1]**
lianli.gao@uestc.edu.cn

**Heng Tao Shen[1,2]**
shenhengtao@hotmail.com

[1]Center for Future Media, University of Electronic Science and Technology of China
[2]Peng Cheng Laboratory

## Abstract

As a crucial approach for compact representation learning, hashing has achieved great success in effectiveness and efficiency. Numerous heuristic Hamming space metric learning objectives are designed to obtain high-quality hash codes. Nevertheless, a theoretical analysis of criteria for learning good hash codes remains largely unexploited. In this paper, we prove that inter-class distinctiveness and intra-class compactness among hash codes determine the lower bound of hash codes' performance. Promoting these two characteristics could lift the bound and improve hash learning. We then propose a surrogate model to fully exploit the above objective by estimating the posterior of hash codes and controlling it, which results in a low-bias optimization. Extensive experiments reveal the effectiveness of the proposed method. By testing on a series of hash-models, we obtain performance improvements among all of them, with an up to $26.5\%$ increase in mean Average Precision and an up to $20.5\%$ increase in accuracy. Our code is publicly available at https://github.com/VL-Group/LBHash.

## 1 Introduction

The explosive increase of multimedia digital content requires people to develop large-scale processing techniques for effective and efficient multimedia understanding [47, 50, 52, 53]. Meanwhile, growing focus on carbon neutrality initiatives urges researchers to handle above issues with low power and memory consumption [30, 31]. Fortunately, hashing, as a key role in compact representation learning, has demonstrated efficiency and effectiveness over the past few decades towards above demands [22]. By digesting multimedia contents into short binary codes, hashing is able to significantly reduce storage, boost speed, and preserve key information [15, 21, 25, 27, 56].

In recent years, learning to hash has shown its advantages in alleviating information loss by performing data-dependent Hamming space projection with a non-linear model [24, 32, 46] compared to hand-crafted hashing functions. The purpose of optimizing hash-models is to generate semantically meaningful hash codes for downstream tasks. A practical convention is to increase similarities among codes of the same category, otherwise the opposite. Such convention is validated to be effective in many applications [8, 26, 33, 40, 41, 48].

Practices in current advances still leave two questions which remain primarily unexploited: When should we determine a hash-model for producing *good* codes, and by what means to achieve this goal? Taking a comprehensive study on the above questions is essential since it not only guides us to

---

*Corresponding author.

36th Conference on Neural Information Processing Systems (NeurIPS 2022).

develop high-performance algorithms in typical hash-based applications but also provides potential possibilities to inspire future works in hash learning.

Although some works [1, 46] try to answer the above questions by building connections between hash codes and common descriptors with domain knowledge, we still seek theoretical guarantees for hash codes' performance. Similar situations occur in related works [30, 44, 45], where objectives to optimize hash-models are designed by intuitions or heuristics.

By formulating correlations among hash codes as inter-class distinctiveness and intra-class compactness, we try to address the above issues and provide a lower bound of hash codes' performance. Based on this, we could further derive from them as objectives to lift the lower bound. Moreover, to effectively train model towards such targets, estimating and further controlling the posterior of hash codes is necessary since bit-level dependence exists in posterior distribution. If we neglect this, bias will be introduced during hash-model optimization, hindering performance. Summarizing both of them, Our main contributions are summarized as follows:

1) To the best of our knowledge, we are the first to formulate a lower bound of hash codes' performance, which is directly proportional to inter-class distinctiveness and intra-class compactness among codes. Based on this, an objective is further proposed to lift this lower bound and thus enhance performance.

2) A novel posterior estimation over hash codes is proposed to perform low-bias optimization towards the above objective. As a non-linear function, it could be utilized for optimizing hash-models via gradient descent. Such plug-and-play component is able to integrate into current hash-models to boost performance.

3) Extensive experiments on three benchmark datasets reveal the effectiveness of the proposed method in two typical tasks. Specifically, we obtain an up to $26.5\%$ increase on mean Average Precision for fast retrieval. We also observe an up to $20.5\%$ increase on accuracy for hash-based recognition.

The rest of paper is organized as follows: We first briefly review current hashing methods and give preliminaries (Secs. 2 and 3). Then, lower bound on hash codes' performance by inter-class distinctiveness and intra-class compactness is proposed (Sec. 4). To fully exploit such guidance, posterior estimation over multivariate Bernoulli distribution is designed for code control (Sec. 5). Experiments (Sec. 6) are then conducted for evaluation.

## 2   Related Works

This paper will focus on data-dependent learning-to-hash methods. Recent advances in building these models are validated to be effective in hash-based recognition and fast retrieval [1, 17, 18, 26, 40]. The typical approach utilizes a projection function to convert raw inputs into compact binary codes and conduct classification or approximate nearest neighbour search on codes for downstream tasks [8, 41, 48]. As mentioned in introduction, to obtain such a projector, current practices suggest that we formulate it as a Hamming space metric learning problem under binary constraints. Pointwise [21, 29, 49], pairwise [4, 5, 25, 26, 37, 55], triplet [10, 28], listwise [43, 54] and prototype-based [2, 6, 17, 51] objectives are all studied for model training, where the core idea is to adjust similarities among samples from same / different categories.

Besides the above objectives, we should also note that hash-model optimization is NP-hard [1, 29, 40, 46]. That is caused by the discrete, binarized hash code formulation. To tackle the issue, previous works propose many methods that target on following purposes: **a)** Surrogate functions that approximate hashing by continuous functions, such as $tanh$ [5], Stochastic neuron [9], $affine$ [38] or Straight-Through [3]. By adopting these, the discrete hash-model optimization is transformed into approximated but easier one via gradient descent. **b)** Non-differentiable optimization algorithm design, which focuses on solutions that directly handle discrete inputs, including Coordinate descent [36], Discrete gradient estimation [13], *etc*. There is also bag of tricks which help for stable training. For example, minimizing quantization error reduces the gap between raw outputs of the model and final hash codes [26], and code-balancing alleviates the problem of producing trivial codes [12].

Please refer to literature reviews on above works [30, 44, 45] for comprehensive studies on learning to hash. We could see two issues exist and remain primarily unexploited in above works. The first is objective design, which is commonly proposed by intuitive and with few theoretical guarantees.

The second is model optimization, where bit-level dependence is essential in code control but most works neglect it. To tackle these, we would firstly give a lower bound of hash codes' performance and utilize it as a guidance for hash-model training. Then, a novel model optimization approach is proposed in a multivariate perspective for low-bias code control.

## 3 Preliminaries

Learning to hash performs data-dependent compact representation learning. Inputs $\boldsymbol{x} \in \boldsymbol{\mathcal{X}} \subseteq \mathbb{R}^d$ are firstly transformed into $h$-dim real-valued vector $\boldsymbol{\ell}$ by $\mathcal{F}_{\boldsymbol{\theta}}$, a projection parameterized by $\boldsymbol{\theta}$: $\mathbb{R}^d \xrightarrow{\mathcal{F}_{\boldsymbol{\theta}}} \mathbb{R}^h$. And $\boldsymbol{\ell}$ is then binarized to $h$ bits binary code $\boldsymbol{b} \in \boldsymbol{\mathcal{B}} \subseteq \mathbb{H}^h$:

$$\boldsymbol{x} \xrightarrow{\mathcal{F}_{\boldsymbol{\theta}}} \boldsymbol{\ell} \xrightarrow{\text{bin}} \boldsymbol{b}, \ where \ \text{bin}\left(\cdot\right) = \begin{cases} +1, & \left(\cdot\right) \geq 0, \\ -1, & \left(\cdot\right) < 0. \end{cases}$$

**Typical usage of hash codes.** Common hashing-based tasks include fast retrieval [2, 19, 27, 39] and recognition [26, 32, 36, 40]. For fast retrieval, given a query under the same distribution of $\boldsymbol{\mathcal{X}}$, we convert it into query hash code $\boldsymbol{q}$ and conduct fast approximate nearest neighbour search in $\boldsymbol{\mathcal{B}}$ to produce rank list $\mathcal{R}$. In $\mathcal{R}$, samples are organized from nearest to furthest to query. Correspondingly among all results in $\mathcal{R}$, true positives $\boldsymbol{tp}^i \in \boldsymbol{TP}$ indicate samples that match with query while others are false positives $\boldsymbol{fp}^i \in \boldsymbol{FP}$ ($i$ indicates rank). As a criterion, Average Precision (AP) is commonly adopted to determine how well the retrieved results are. It encourages true positives to have higher ranks than false positives: $AP = \frac{1}{|\boldsymbol{TP}|} \sum_{\forall \boldsymbol{tp}^{i,m} \in \boldsymbol{TP}} P@i$, where $P@i$ is the rank-$i$ precision. As for hash-based recognition, since current works formulate it as a variant of retrieval, we could still adopt AP as a criterion to indicate performance. The detailed explanation is placed in Supp. Sec. C.

## 4 Lower Bound by Inter-Class Distinctiveness and Intra-Class Compactness

We start at an arbitrary rank list for studying the characteristics of codes in the above scenarios. To calculate AP easily, we first introduce *mis-rank* that indicates how many false positives take higher places than true positives. We refer readers to Supp. Secs. A and B for detailed description and proof.

**Definition.** Mis-rank $m$ indicates how many false positives have higher ranks than a true positive, *e.g.*, $\boldsymbol{tp}^{i,m}$ is at rank $i$, meanwhile $m = |\{d\left(\boldsymbol{q}, \boldsymbol{fp}\right) < d\left(\boldsymbol{q}, \boldsymbol{tp}^{i,m}\right)\}|$, where $d\left(\boldsymbol{q}, \cdot\right)$ is distance between $\boldsymbol{q}$ and a sample, and $|\cdot|$ is number of elements in set.

*Remark.* The rank-$i$ precision $P@i$ is derived to be $(i-m)/i$.

Then, average precision could be derived as:

$$AP = \frac{1}{|\boldsymbol{TP}|} \sum_{\forall \boldsymbol{tp}^{i,m} \in \boldsymbol{TP}} P@i = \frac{1}{|\boldsymbol{TP}|} \sum_{\forall \boldsymbol{tp}^{i,m} \in \boldsymbol{TP}} \frac{i - m}{i},$$

for all true positives in rank list. This is because the rank-$i$ precision $P@i$ for any true positive $\boldsymbol{tp}^{i,m}$ equals to $(i-m)/i$.

From the derivation, we could immediately obtain that $AP$ increases *iff* $m$ decreases. Therefore, now we could focus on how to reduce $m$ to thereby increase average precision. Noticed that the value of $m$ is highly related to two distances, $d\left(\boldsymbol{q}, \boldsymbol{fp}\right)$ and $d\left(\boldsymbol{q}, \boldsymbol{tp}\right)$, the following proposition is raised:

**Proposition.**

$$\overline{m} \propto \frac{\max d\left(\boldsymbol{q}, \boldsymbol{tp}\right)}{\min d\left(\boldsymbol{q}, \boldsymbol{fp}\right)} \ \forall \boldsymbol{tp} \in \boldsymbol{TP}, \ \boldsymbol{fp} \in \boldsymbol{FP}$$

where $\bar{\cdot}$ denotes upper bound. Correspondingly,

$$\underline{AP} \propto \frac{\min d\left(\boldsymbol{q}, \boldsymbol{fp}\right)}{\max d\left(\boldsymbol{q}, \boldsymbol{tp}\right)} \ \forall \boldsymbol{tp} \in \boldsymbol{TP}, \ \boldsymbol{fp} \in \boldsymbol{FP}$$

where $\underline{\cdot}$ denotes lower bound.

If inputs $\boldsymbol{x}^j$ have associated labels $y^j$, the above studies are able to be further generalized. Class-wise centers, inter-class distinctiveness and intra-class compactness are introduced to formulate our final lower-bound of hash codes' performance.

**Definition.** Center $\boldsymbol{c}^c \in \boldsymbol{C}$ is a representative hash code of a specific class $c$:

$$\boldsymbol{c}^c = \arg\min_{\boldsymbol{b}} \sum_j d\left(\boldsymbol{b}, \boldsymbol{b}^j\right), \ \forall \left\{\boldsymbol{b}^j \mid y^j = c\right\}$$

where $\boldsymbol{b}^j$ is the $j$-th sample in set $\boldsymbol{\mathcal{B}}$ and $y^j$ is the label of $\boldsymbol{b}^j$.

**Definition.** *Inter-class distinctiveness* is $\min \mathcal{D}_{inter}$ where $\mathcal{D}_{inter}$ is a set that measures distances between all centers over $\boldsymbol{C}$: $\left\{d\left(\boldsymbol{c}^j, \boldsymbol{c}^k\right) \mid \forall \boldsymbol{c}^j, \boldsymbol{c}^k \in \boldsymbol{C}\right\}$. In contrast, *intra-class compactness* is $1/\max \mathcal{D}_{intra}$, where $\mathcal{D}_{intra} = \left\{d\left(\boldsymbol{b}^j, \boldsymbol{c}^c\right) \mid \forall y^j = c\right\}$ measures the distances among samples of the same class to their corresponding center.

Combining above propositions and definitions, we could reach:

**Proposition.**

$$\underline{AP} \propto \frac{\min d\left(\boldsymbol{q}, \boldsymbol{fp}\right)}{\max d\left(\boldsymbol{q}, \boldsymbol{tp}\right)} \geq const \cdot \frac{\min \mathcal{D}_{inter}}{\max \mathcal{D}_{intra}}. \tag{1}$$

This proposition reveals how AP is affected by the above two inter- and intra- factors. Actually, it could cover common scenarios as a criterion of hash codes' performance, *e.g.*, fast retrieval and recognition, which will be explained in supplementary materials. Intuitively, lifting this lower bound would enhance performance, formulating the following objective:

$$maximize \ \min \mathcal{D}_{inter}, \tag{2}$$

$$minimize \ \max \mathcal{D}_{intra}, \tag{3}$$

$$s.t. \ \boldsymbol{\mathcal{X}} \subseteq \mathbb{R}^d \xrightarrow{\mathcal{F}_{\boldsymbol{\theta}}, \ \mathrm{bin}} \boldsymbol{\mathcal{B}} \subseteq \mathbb{H}^h.$$

Inspired by [51], we give a specific solution to implement Eqs. (2) and (3) under supervised hashing by firstly defining distance maximized class-specific centers and then shrinking hash codes to their corresponding centers. We will validate the effectiveness of such a solution in experiments. It is worth noting that achieving above goal is an open topic, including unsupervised or other complex scenarios, which we leave for future study. Next, to optimize hash-models with the above supervision, we introduce posterior estimation over hash codes and then control it.

## 5 Posterior Estimation for Code Control

It is essentially hard to minimize the distance between a hash code and a target to achieve Eq. (3) for two reasons. Firstly, the posterior distribution of $\boldsymbol{\mathcal{B}}$ given inputs $\boldsymbol{\mathcal{X}}$ is formulated to be under multivariate Bernoulli distribution: $p\left(\boldsymbol{\mathcal{B}} \mid \boldsymbol{\mathcal{X}}\right) \sim \mathrm{MVB}$. If we do not take correlations among different variables into account, optimization on $\boldsymbol{\mathcal{B}}$ is biased. Secondly, dealing with hash codes is a $\{-1, +1\}$ discrete optimization, which is not trivial to handle. Therefore, we try to organize Eq. (3) as a Maximum Likelihood Estimation (MLE) to tackle the above issues. Considering the definition of Hamming distance between an arbitrary hash code $\boldsymbol{b}$ and its target $\boldsymbol{t}$:

$$d\left(\boldsymbol{b}, \boldsymbol{t}\right) = \sum_i^h \mathbb{1}\left\{\boldsymbol{b}_i \neq \boldsymbol{t}_i\right\} = \sum_i^h \mathbb{1}\left\{\boldsymbol{\ell}_i \boldsymbol{t}_i < 0\right\}$$

where $\mathbb{1}$ is the characteristic function and $(\cdot)_i$ indicates value on the $i$-th dimension. Since Hamming distance measures how many bits are different between two codes, the probability of $d\left(\boldsymbol{b}, \boldsymbol{t}\right) = \delta$ can be formulated as:

$$p\left(d\left(\boldsymbol{b}, \boldsymbol{t}\right) = \delta\right) = \sum_{\forall i \in \binom{h}{\delta}, \ j \notin \binom{h}{\delta}} \left\{p\left(\boldsymbol{b}_i \neq \boldsymbol{t}_i, \ \boldsymbol{b}_j = \boldsymbol{t}_j\right)\right\}.$$

Therefore, $\boldsymbol{b}$ and $\boldsymbol{t}$ have distance $\delta$ *iff* $p\left(d\left(\boldsymbol{b}, \boldsymbol{t}\right) = \delta\right)$ is maximized. However, to precisely calculate it is difficult, since it involves the joint probability of $\boldsymbol{b}$. Therefore, we try to estimate all joint probabilities by adopting a surrogate model $\mathcal{P}_{\boldsymbol{\pi}}$ parameterized by $\boldsymbol{\pi}$ to perform estimation:

$$\boldsymbol{\ell} \xrightarrow{\mathcal{P}_{\boldsymbol{\pi}}} \boldsymbol{o}, \ where \ \boldsymbol{o} \in \boldsymbol{O} \subseteq \mathbb{R}^{2^h} \tag{4}$$

where $o$ is the probabilities of a Categorical distribution $p\left(\boldsymbol{O} \mid \boldsymbol{\mathcal{X}}\right) \widehat{=} p\left(\boldsymbol{\mathcal{B}} \mid \boldsymbol{\mathcal{X}}\right)$, which contains $2^h$ entries. Each entry of $\boldsymbol{o}$ estimates one of a specific joint probability by feeding $\boldsymbol{\ell}$ into $\mathcal{P}_{\boldsymbol{\pi}}$:

$$\boldsymbol{o}_i \widehat{=} p\left(\left(\boldsymbol{b}_k > 0\right)^{\mathbb{1}\{\boldsymbol{\iota}_k=1\}}, \left(\boldsymbol{b}_k < 0\right)^{\mathbb{1}\{\boldsymbol{\iota}_k=0\}}\right), \; \begin{array}{l} 1 \leq k \leq h, \\ 0 \leq i \leq 2^h-1 \end{array}$$

where $\boldsymbol{\iota}$ is the $h$ bits binary representation of $i$. For example, $\boldsymbol{o}_3 = p\left(\boldsymbol{b} = (\text{-1-1+1+1})\right)$ when $h = 4$. To train $\mathcal{P}_{\boldsymbol{\pi}}$, we perform Maximum Likelihood Estimation (MLE) over $p\left(\boldsymbol{O} \mid \boldsymbol{\mathcal{X}}\right)$:

$$L_{\boldsymbol{\pi}}\left(\mathcal{P}_{\boldsymbol{\pi}}; \boldsymbol{o}\right) = -\log \boldsymbol{o}_i,$$
$$where \; i = \sum_{k=1}^{h} \mathbb{1}\left\{\boldsymbol{b}_k > 0\right\} \cdot 2^{k-1}. \tag{5}$$

If $\mathcal{P}_{\boldsymbol{\pi}}$ is ready for estimation, we could directly adopt it as a nice non-linear function to maximize $p\left(d\left(\boldsymbol{b}, \boldsymbol{t}\right) = k\right)$. Note that to realize Eq. (3), we want $\boldsymbol{b}$ and its corresponding center $\boldsymbol{c}$ are closest, *i.e.* maximize $p\left(d\left(\boldsymbol{b}, \boldsymbol{c}\right) = 0\right)$. So, we could reformulate this maximization as another MLE:

$$L_{\boldsymbol{\theta}}\left(\mathcal{F}_{\boldsymbol{\theta}}; \boldsymbol{b}\right) = -\log p\left(d\left(\boldsymbol{b}, \boldsymbol{c}\right) = 0\right),$$
$$where \; p\left(d\left(\boldsymbol{b}, \boldsymbol{c}\right) = 0\right) = p\left(\boldsymbol{b}_i = \boldsymbol{c}_i, 1 \leq i \leq h\right), \tag{6}$$

which could be optimized by calculating surrogate gradients of Eq. (6) through $\mathcal{P}_{\boldsymbol{\pi}}$:

$$\frac{\partial \hat{L}_{\boldsymbol{\theta}}}{\partial \boldsymbol{\theta}} = \frac{\partial \left\{-\log \boldsymbol{o}_i; \; i = \sum_{k=1}^{h} \mathbb{1}\left\{\boldsymbol{c}_k > 0\right\} \cdot 2^{k-1}\right\}}{\partial \boldsymbol{\pi}} \frac{\partial \boldsymbol{\pi}}{\partial \boldsymbol{\ell}} \frac{\partial \boldsymbol{\ell}}{\partial \boldsymbol{\theta}} \widehat{=} \frac{\partial L_{\boldsymbol{\theta}}}{\partial \boldsymbol{\theta}}. \tag{7}$$

Unfortunately, such joint probability estimation on MVB requires $\mathcal{O}\left(2^h\right)$ complexity, which is not flexible when $h$ is large. By adopting the idea of block code [42], the above estimation is able to perform on long hash bits by separating hash codes into a series of blocks. Specifically, any $h$ bits hash codes can be split into $u$ blocks while each block consumes $h/u$ bits. Correspondingly, $u$ independent surrogate networks $\mathcal{P}_{\boldsymbol{\pi}_i}, 1 \leq i \leq u$ are adopted to perform the above estimation and back-propagation simultaneously. With such decomposition, the whole problem is transformed into $u$ sub-problems with a reduced complexity $\mathcal{O}\left(2^h\right) \rightarrow \mathcal{O}\left(u \cdot 2^{h/u}\right)$.

As a summarization all of things, overall optimization via gradient descent is placed in Alg. 1. We first perform Eq. (2) step by using the pre-defined centers and then perform Eq. (3) step by back-propagation via Eq. (7). Two models $\mathcal{F}_{\boldsymbol{\theta}}, \mathcal{P}_{\boldsymbol{\pi}}$ are optimized with learning rate $\eta_1, \eta_2$ respectively.

---

**Algorithm 1** One of implementations under supervised circumstance.

---
1: **procedure** TRAIN($\mathcal{F}_{\boldsymbol{\theta}}$, $\mathcal{P}_{\boldsymbol{\pi}}$)                     ▷ Training procedure of two models $\mathcal{F}_{\boldsymbol{\theta}}$, $\mathcal{P}_{\boldsymbol{\pi}}$.
2:     Generate class-specific centers $\boldsymbol{c} \in C, |C| = $ Class-num;                     ▷ (Eq. (2)).
3:     **repeat**                                                                          ▷ Main training loop.
4:         Sample $\boldsymbol{x}$ from $\boldsymbol{\mathcal{X}}$ with label $y$;
5:         $\boldsymbol{\ell} = \mathcal{F}_{\boldsymbol{\theta}}\left(\boldsymbol{x}\right)$;
6:         $\boldsymbol{o} = \mathcal{P}_{\boldsymbol{\pi}}\left(\boldsymbol{\ell}\right)$;
7:         $\boldsymbol{\pi} \leftarrow \boldsymbol{\pi} - \eta_1 \frac{\partial L_{\boldsymbol{\pi}}}{\partial \boldsymbol{\pi}}$;                                           ▷ (Eq. (5)).
8:         $\boldsymbol{\theta} \leftarrow \boldsymbol{\theta} - \eta_2 \frac{\partial \hat{L}_{\boldsymbol{\theta}}}{\partial \boldsymbol{\theta}}$ with corresponding center $\boldsymbol{c}$;             ▷ Eq. (3), Eq. (7).
9:     **until** Total epoch exceeds;
10:     **return** $\mathcal{F}_{\boldsymbol{\theta}}$;                                                     ▷ Optimized hash-model $\mathcal{F}_{\boldsymbol{\theta}}$
11: **end procedure**

---

## 6 Experiments

We conduct extensive experiments on three benchmark datasets to confirm the effectiveness of our proposed method. To make fair comparisons, we first provide experiments setup.

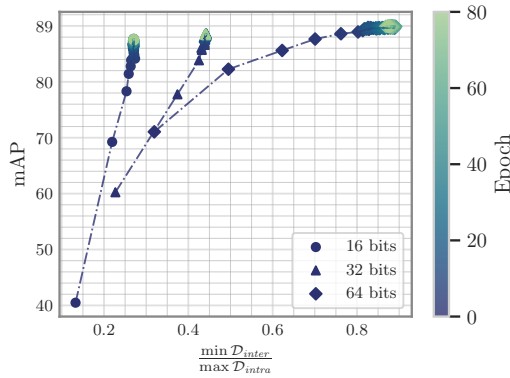

Figure 1: mAP *w.r.t.* $\frac{\min \mathcal{D}_{inter}}{\max \mathcal{D}_{intra}}$ per epoch on ImageNet $64$ bits during training. These two values have a positive relationship.

Figure 2: Convergence curve of three variants on ImageNet $64$ bits. Our $\mathcal{P}_{\pi}$ helps for achieving higher performance than other two.

## 6.1 Setup

Our experiments focus on performance comparisons on two typical hash-based tasks, fast retrieval and recognition, with and without integrating our proposed method. The general evaluation pipeline is to train models in the training split, then hash all samples in the base split. Then for retrieval, any hashed queries in query split are used to get rank lists of the base split from nearest to farthest according to Hamming distance. As for recognition, we adopt a $k$NN classifier or a linear model to produce queries' predictions.

### 6.1.1 Datasets

Our experiments are conducted on three datasets, varying in scale, variety and perplexity to validate methods' performance in different scenarios. Both single-label and multi-label datasets are included. We follow previous works to generate three splits, which are detailed below:

**CIFAR-10 [20]** is a single-label 10-class dataset. The whole dataset contains $6,000$ images for each class. Following [5], we split the dataset into $500 \mid 5,400 \mid 100$ for each class randomly as train, base and query splits, respectively.

Table 1: Total training time with different variants, where we substitute our surrogate model with BCE loss from [51] or Cauchy loss from [4].

| Method | Training time per epoch (s) | | |
|---|---|---|---|
| | 16 bits | 32 bits | 64 bits |
| BCE | 38.13 | 38.20 | 38.52 |
| Cauchy | 38.21 | 38.30 | 38.42 |
| Ours | 38.95 | 39.78 | 40.84 |

**NUS-WIDE [7]** consists of $81$ labels and images may have one or more labels. We follow previous works [5] to pick the most frequent 21 labels and their associated images $(195,834)$ for experiments. Specifically, $193,734$ images are randomly picked to form the base split while remaining $2,100$ images are adopted for queries. $10,500$ images are randomly sampled from the base split for training models.

**ImageNet [11]** is a large-scale dataset consists of $1,000$ classes. To conduct experiments, we follow [5] to pick a subset of 100 classes where all images of these classes in the training set / validation set are as base split / query split respectively $(128,503 \mid 4,983$ images). We then randomly sample 100 images per class in the base split for training.

### 6.1.2 Implementation Details

Our method is able to integrate into common hash-models. Due to the limitation of computation resources, we choose a series of representative deep hashing methods for comparison, including **HashNet [5]**, **DBDH [55]**, **DSDH [26]**, **DCH [4]**, **GreedHash [40]** and **CSQ [51]**. When our method is integrated, we append a subscript $(\cdot)_D$ to the original methods' name. All methods adopted for

experiments are implemented from a public benchmark with PyTorch [34].[2] For fair comparisons, we conduct experiments with the same backbone (ResNet-50 [16]) and hyper-parameters for all tested methods. We adopt Adam [35] optimizer with default configuration and learning rate of our method $\eta_1 = \eta_2 = 1e^{-3}$ for training. For multi-label datasets, we simply modify Alg. 1 Line 8 with the sum of multiple losses. Block number $u$ of $\mathcal{P}$ is set to $^{bits}/_8$. For example, if the length of hash code is 64, there will be 8 sub-models $\mathcal{P}_{\pi_1} \sim \mathcal{P}_{\pi_8}$ trained in parallel. To mitigate randomness, we report average performance for 5 runs on all experiments. The evaluation metric adopted for retrieval is mean Average Precision (mAP@$R$) where $R = 54,000 \mid 5,000 \mid 1,000$ on CIFAR-10, NUS-WIDE, ImageNet, respectively. For recognition performance, if methods have an auxiliary classification branch, we directly use its predictions from it. Otherwise, we adopt a $k$NN classifier built on base split and vote with 100 nearest neighbours.

## 6.2 Ablation Study

In the ablation study, we try to reveal the correctness of our proposed objectives Eqs. (2) and (3), and the effectiveness of our proposed posterior estimation model $\mathcal{P}_{\pi}$. We answer the following questions by conducting experiments on ImageNet to demystify the above concerns. We focus on retrieval performance in the ablation study, while we observe similar results on recognition.

**Is hash-model's performance lower-bounded by Eq. (1)?** Validating correctness of Eq. (1) is important, but it is essentially hard to conduct experiments to confirm it. Nevertheless, we still reveal the correlation between model performance and inter-class distinctiveness / intra-class compactness by tracking mAP *w.r.t.* $\frac{\min \mathcal{D}_{inter}}{\max \mathcal{D}_{intra}}$ during training. To calculate $\mathcal{D}_{inter}$ and $\mathcal{D}_{intra}$, we first hash all samples from the base split and calculate their centers of them over all classes. The 99.9 percentile of $\min \mathcal{D}_{inter}$ and $\max \mathcal{D}_{intra}$ is picked to avoid outliers. We conduct tracking on CSQ$_D$ with 16, 32, 64 bits per epoch to draw Fig. 1. As the figure shows, lines go from lower left to upper right with small fluctuations, which indicates the linear relationship between mAP and $\frac{\min \mathcal{D}_{inter}}{\max \mathcal{D}_{intra}}$ on all bit-lengths. This may partially confirm our theoretical analysis in Sec. 4.

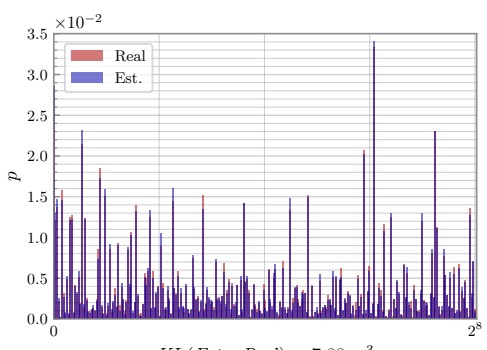

$$KL\,(Est.,\ Real) = 7.88e^{-3}$$

| Method | $KL_{real}$ |
|--------|-------------|
| Naïve  | 0.538 |
| Ours   | 0.008 |

Figure 3: Multivariate Bernoulli estimation by our proposed $\mathcal{P}_{\pi}$, and KL-divergence comparisons with naïve independent estimation.

**Convergence speed and efficiency of $\mathcal{P}_{\pi}$.** To verify the convergence speed and efficiency of our posterior estimation model $\mathcal{P}_{\pi}$, we test it by substituting with two variants: BCE from [51] and Cauchy loss from [4]. mAP is evaluated per epoch for three variants with three bit-lengths 16, 32, 64 and the convergence curve is plotted in Fig. 2. From the figure, we could see that ours and BCE's convergence speed are similar, while Cauchy is slightly slow. Meanwhile, our method has continuously higher performance than other two after $\sim 10$ epochs for all bit-lengths, which is potentially due to our method could perform low-bias optimization. Note that when training with Cauchy loss, performance is instead lower when codes are longer. This may be caused by the inner-product based Cauchy loss could not handle long bits. We also measure averaged training time per epoch of all variants on a single NVIDIA RTX 3090, which is placed in Tab. 1. In this table, our model does not consume significant longer time than others to train. The above observations reveal the efficiency of our proposed method. It could be a drop-in replacement without introducing significant overhead. Moreover, convergence speed and efficiency will not drop when the bit-length becomes long, which reveals the flexibility of the blocked code design.

**Could $\mathcal{P}_{\pi}$ estimate multivariate Bernoulli distribution?** We design an extra toy experiments to validate the ability of estimating multivariate Bernoulli distribution by our surrogate model $\mathcal{P}_{\pi}$.

---

[2] https://github.com/swuxyj/DeepHash-pytorch

Table 2: mAP comparisons on three benchmark datasets for $16, 32, 64$ bits codes. $_{\uparrow(\cdot)}$ indicates performance enhancement with our method integrated.

| Method | CIFAR-10 | | | NUS-WIDE | | | ImageNet | | |
|---|---|---|---|---|---|---|---|---|---|
| | 16 bits | 32 bits | 64 bits | 16 bits | 32 bits | 64 bits | 16 bits | 32 bits | 64 bits |
| HashNet | 63.0 | 81.5 | 84.7 | 77.8 | 83.3 | 85.0 | 50.6 | 63.1 | 68.4 |
| HashNet$_D$ | $79.1_{\uparrow16.1}$ | $82.3_{\uparrow0.8}$ | $85.3_{\uparrow0.6}$ | $78.6_{\uparrow0.8}$ | $84.0_{\uparrow0.7}$ | $86.2_{\uparrow1.2}$ | $72.6_{\uparrow22.0}$ | $84.3_{\uparrow21.2}$ | $87.6_{\uparrow19.2}$ |
| DBDH | 83.1 | 85.0 | 85.6 | 82.8 | 84.6 | 85.7 | 55.7 | 63.8 | 76.9 |
| DBDH$_D$ | $84.6_{\uparrow1.5}$ | $86.0_{\uparrow1.0}$ | $86.5_{\uparrow0.9}$ | $83.8_{\uparrow1.0}$ | $85.6_{\uparrow1.0}$ | $86.5_{\uparrow0.8}$ | $82.2_{\uparrow26.5}$ | $85.8_{\uparrow22.0}$ | $86.1_{\uparrow9.2}$ |
| DSDH | 75.6 | 83.1 | 84.5 | 83.3 | 84.5 | 85.6 | 57.2 | 72.1 | 75.3 |
| DSDH$_D$ | $84.3_{\uparrow8.7}$ | $84.6_{\uparrow1.5}$ | $87.3_{\uparrow2.8}$ | $83.6_{\uparrow0.3}$ | $85.3_{\uparrow0.8}$ | $86.4_{\uparrow0.8}$ | $81.7_{\uparrow24.5}$ | $87.3_{\uparrow15.2}$ | $87.9_{\uparrow12.6}$ |
| DCH | 83.4 | 84.4 | 85.3 | 80.7 | 81.7 | 80.9 | 85.5 | 86.2 | 86.4 |
| DCH$_D$ | $83.6_{\uparrow0.2}$ | $84.6_{\uparrow0.2}$ | $87.1_{\uparrow1.8}$ | $82.9_{\uparrow2.2}$ | $84.2_{\uparrow2.5}$ | $84.9_{\uparrow4.0}$ | $86.1_{\uparrow0.6}$ | $87.5_{\uparrow1.3}$ | $88.1_{\uparrow1.7}$ |
| GreedHash | 83.3 | 84.3 | 86.9 | 78.6 | 80.3 | 82.0 | 83.1 | 85.9 | 86.4 |
| GreedHash$_D$ | $85.2_{\uparrow1.9}$ | $85.5_{\uparrow1.2}$ | $87.6_{\uparrow0.7}$ | $79.4_{\uparrow0.8}$ | $83.1_{\uparrow2.8}$ | $85.7_{\uparrow3.7}$ | $83.8_{\uparrow0.7}$ | $86.6_{\uparrow0.7}$ | $86.8_{\uparrow0.4}$ |
| CSQ | 83.2 | 83.4 | 84.7 | 82.0 | 83.5 | 84.6 | 83.4 | 86.9 | 87.9 |
| CSQ$_D$ | $88.7_{\uparrow5.5}$ | $89.2_{\uparrow5.8}$ | $90.3_{\uparrow5.6}$ | $83.3_{\uparrow1.3}$ | $85.3_{\uparrow1.8}$ | $85.8_{\uparrow1.2}$ | $88.5_{\uparrow5.1}$ | $89.5_{\uparrow2.6}$ | $90.2_{\uparrow2.3}$ |

Specifically, 8 bits MVB is generated with randomly 256 joint probabilities. We then take $10,000$ samples from it as inputs to train $\mathcal{P}_\pi$. The model will further estimate distribution by feeding another 100 samples and taking the mean of all $o$s (Eq. (4)) as result. The estimated distribution is evaluated by Kullback–Leibler divergence [23] ($KL$) with real distribution, as well as bar plot, demonstrated in Fig. 3. As the figure shows, our predicted joint probabilities almost cover real probabilities and $KL$ is low. As a comparison, if we estimate real distribution with the product of edge probabilities directly ($w/o$ correlations between variables), $KL$ will be significantly increased (row 1 in table). Such toy experiment reveals that our method is better for estimating MVBs than the naïve one.

## 6.3 Performance Enhancements when Integrating into State-of-the-Art

To evaluate performance gain when integrating our proposed method into hash-models, we conduct experiments on three datasets. Specifically, we first report the original performance of tested models and then re-run with our method incorporated to make a comparison. Both $16, 32, 64$ bits results are reported to show performance from short codes to long codes.

**Retrieval Performance.** Retrieval performance comparisons under mAP is shown in Tab. 2. With our integrated method, all tested hashing methods have performance enhancement. We observe an up to $26.5\%$ increase and $5.02\%$ on average, which is a significant improvement. Specifically, performance increase on HashNet, DBDH and DSDH is higher than other methods (first 3 rows), especially on ImageNet dataset. A potential reason is that all of these three methods use pairwise metric learning objectives and inner-product to approximate Hamming distance, which may not handle optimization well when dataset size and class number is large. Meanwhile, our method works on multi-label datasets where all methods also obtain perfor-

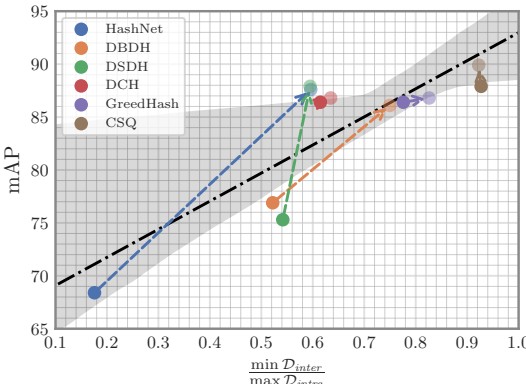

Figure 4: mAP $w.r.t.$ $\frac{\min \mathcal{D}_{inter}}{\max \mathcal{D}_{intra}}$ for different methods. By integrating our auxiliary objective, we could observe significant performance and $\frac{\min \mathcal{D}_{inter}}{\max \mathcal{D}_{intra}}$ increase in most cases (from solid dots to translucent dots). Regression line is also plotted with $95\%$ confidence interval to reveal linear relationship between two metrics.

mance gain, indicating that multi-label data may also benefit from our objective. Furthermore, we

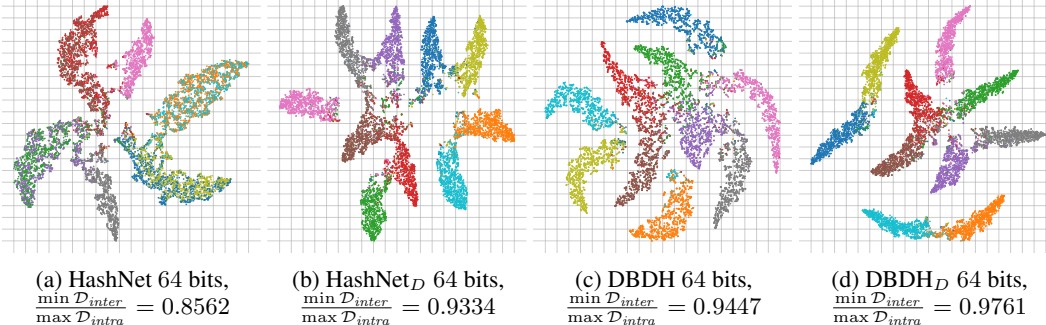

(a) HashNet 64 bits, $\frac{\min \mathcal{D}_{inter}}{\max \mathcal{D}_{intra}} = 0.8562$    (b) HashNet$_D$ 64 bits, $\frac{\min \mathcal{D}_{inter}}{\max \mathcal{D}_{intra}} = 0.9334$    (c) DBDH 64 bits, $\frac{\min \mathcal{D}_{inter}}{\max \mathcal{D}_{intra}} = 0.9447$    (d) DBDH$_D$ 64 bits, $\frac{\min \mathcal{D}_{inter}}{\max \mathcal{D}_{intra}} = 0.9761$

Figure 5: t-SNE visualization for HashNet and DBDH $w/$ and $w/o$ our integration, where the $(\cdot)_D$ variants have more compact structures and larger margin among different classes than original ones.

also observe a $3.46\%$ average mAP increase on CSQ. It adopts a similar objective but a different optimization approach with ours. This increase shows the effectiveness of our proposed posterior estimation approach, which could perform better optimization than theirs.

**Recognition Performance.** Similarly, we report the recognition performance of these methods by measuring classification accuracy on ImageNet with $16, 32, 64$ bits codes. Results are reported in Tab. 3. Generally, we observe similar performance gains. When methods have our integration, their accuracy is significantly increased, up to $20.5\%$ and $5.3\%$ on average. Specifically, some of them utilize an auxiliary classification branch to produce predictions, *i.e.*, GreedyHash and DSDH. While for others, we adopt a $k$NN classifier to classify samples by voting with the base split. We obtain increase for both approaches. This indicates that our approach also helps for hash-based recognition in different scenarios, by leveraging inter-class distinctiveness and intra-class compactness.

From the above comparisons, we confirm the effectiveness of our proposed method. Overall, our method is validated to be flexible and promising to deploy in the above scenarios with various bits.

Table 3: Recognition performance on ImageNet.

| Method | ImageNet | | |
|---|---|---|---|
| | 16 bits | 32 bits | 64 bits |
| HashNet | 69.1 | 74.1 | 82.4 |
| HashNet$_D$ | **83.2**$_{\uparrow 14.1}$ | **88.1**$_{\uparrow 14.0}$ | **89.5**$_{\uparrow 7.1}$ |
| DBDH | 80.9 | 82.9 | 88.2 |
| DBDH$_D$ | **86.8**$_{\uparrow 5.9}$ | **89.1**$_{\uparrow 6.2}$ | **88.4**$_{\uparrow 0.2}$ |
| DSDH | 40.6 | 49.9 | 54.1 |
| DSDH$_D$ | **61.1**$_{\uparrow 20.5}$ | **62.1**$_{\uparrow 12.2}$ | **63.2**$_{\uparrow 9.1}$ |
| DCH | 88.5 | 89.2 | 88.1 |
| DCH$_D$ | **88.9**$_{\uparrow 0.4}$ | **89.5**$_{\uparrow 0.3}$ | **88.7**$_{\uparrow 0.6}$ |
| GreedHash | 79.1 | 86.8 | 87.0 |
| GreedHash$_D$ | **80.2**$_{\uparrow 1.1}$ | **87.1**$_{\uparrow 0.3}$ | **88.4**$_{\uparrow 1.4}$ |
| CSQ | 87.7 | 89.0 | 89.4 |
| CSQ$_D$ | **87.9**$_{\uparrow 0.2}$ | **89.1**$_{\uparrow 0.1}$ | **91.0**$_{\uparrow 1.6}$ |

### 6.4 Qualitative Comparisons

**t-SNE visualization.** To further analyze how our objective affects hash codes by adjusting $\mathcal{D}_{inter}$ and $\mathcal{D}_{intra}$, we conduct codes visualization by t-SNE [14] on HashNet and DBDH. Specifically, we randomly sample $5,000$ 64 bits codes from CIFAR-10 base split. t-SNE is then performed on these codes as shown in Fig. 5, where points' color indicates class. From these figures, we give our humble explanation. Firstly, please look at Fig. 5(a), codes extracted from the original HashNet are mixed among different classes. While in Fig. 5(c), clusters are loose. These representations result in low $\min \mathcal{D}_{inter}$ and high $\max \mathcal{D}_{intra}$. When they are trained along with our method (Figs. 5(b) and 5(d)), we get more separable and distinct clusters compared to the original ones. Meanwhile, codes in a cluster are more compact than the original methods. Quantitative results of $\frac{\min \mathcal{D}_{inter}}{\max \mathcal{D}_{intra}}$ placed under figures also reveal this phenomenon.

**mAP *w.r.t.* $\frac{\min \mathcal{D}_{inter}}{\max \mathcal{D}_{intra}}$ visualization.** To further confirm the relationship between hashing performance and $\frac{\min \mathcal{D}_{inter}}{\max \mathcal{D}_{intra}}$, as in Sec. 6.2, we plot mAP *w.r.t.* $\frac{\min \mathcal{D}_{inter}}{\max \mathcal{D}_{intra}}$ for all tested methods on 64 bits ImageNet base split (Fig. 4). As the figure shows, most of the methods obtain higher mAP with larger $\frac{\min \mathcal{D}_{inter}}{\max \mathcal{D}_{intra}}$

and move from lower-left region to upper-right region (solid dots to translucent dots indicate original methods to integrated methods). Meanwhile, according to the positions of these dots, mAP and $\frac{\min \mathcal{D}_{inter}}{\max \mathcal{D}_{intra}}$ are supposed to be under linear relationship for different methods, *i.e.* higher $\frac{\min \mathcal{D}_{inter}}{\max \mathcal{D}_{intra}}$ leads to higher mAP. Therefore, we give a regression line with $95\%$ confidence interval on the plot to confirm our observations. Extended from this, our proposed lower bound could also be a criterion of hash codes' performance.

## 7  Conclusion

In this paper, we conduct a comprehensive study on the characteristics of hash codes. As a result, we prove that hash codes' performance is lower-bounded by inter-class distinctiveness and intra-class compactness. Formulating such a lower bound as an objective, we could further lift it in hash-model training. Meanwhile, our proposed surrogate model for posterior estimation over hash codes' fully exploits the above guidance to perform low-bias model optimization and finally produce good codes. Extensive experiments conducted on three benchmark datasets confirm the effectiveness of our proposed method. We are able to boost current hash-models' performance with a flexible integration.

## Limitation and Broader Impacts

Hashing and related compact representation learning are able to significantly reduce computational requirements while improving memory efficiency when deploying to real scenarios. However, the main challenge for these techniques is information loss which results in a performance drop. To tackle this, our work on on hash codes' performance gives a lower bound, validated in two specific tasks. However, generalizing such lower bounds to various scenarios, *e.g.* semi-supervised, unsupervised hash learning, are left for future study. Meanwhile, our posterior estimation approach is not verified for extremely long bits. Nevertheless, our work may still provide the potential to inspire researchers to improve not only hash learning, but also broader areas that adopt metric learning. Our study on posterior estimation may also help for precise discrete optimizations.

## Acknowledgments and Disclosure of Funding

This work is supported by National Key Research and Development Program of China (No. 2018AAA0102200), the National Natural Science Foundation of China (Grant No. 62122018, No. 61772116, No. 62020106008, No. 61872064).

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
