# Appendix: A Lower Bound of Hash Codes' Performance

**Xiaosu Zhu[1]**
xiaosu.zhu@outlook.com

**Jingkuan Song[1]\***
jingkuan.song@gmail.com

**Yu Lei[1]**
leiyu6969@gmail.com

**Lianli Gao[1]**
lianli.gao@uestc.edu.cn

**Heng Tao Shen[1,2]**
shenhengtao@hotmail.com

[1]Center for Future Media, University of Electronic Science and Technology of China
[2]Peng Cheng Laboratory

In this supplementary material, we discuss the following topics: Firstly, we give explanation in Appendix A to demystify concepts of rank lists. Then, proofs of the propositions in main paper are given in Appendix B. We further discuss why to adopt $AP$ as a criterion of hash codes' performance in Appendix C. To train hash-models, we treat the posterior of hash codes to be under the multivariate Bernoulli distribution. We explain why and how to perform posterior estimation in Appendices D and E. Additional experiments are finally given in Appendix F.

## A   Definitions

**The queries, true positives and false positives.**   We demonstrate concepts of queries, true positives and false positives in rank lists in Figs. 1 and 2(a) for easy understanding. As the figure shows, any true positives or false positives are assigned with ranks $i$. Meanwhile, any true positives are also tagged by mis-ranks $m$ introduced in this paper. $m$ indicates how many false positives have the higher ranks than the current true positive.

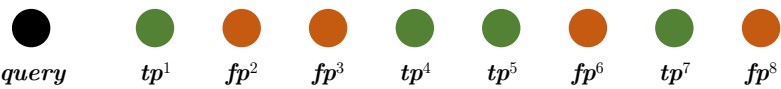

Figure 1: An example rank list for demonstration. True positives are green while false positives are orange. All positives have ranks $i$ placed on upper-right.

We assume distances between query and any positive samples are different with each other. Then, we obtain following inequalities according to the property of a rank list:

$$d\left(\boldsymbol{q}, \boldsymbol{tp}^1\right) < d\left(\boldsymbol{q}, \boldsymbol{fp}^2\right) < \cdots < d\left(\boldsymbol{q}, \boldsymbol{fp}^8\right). \tag{1}$$

---

\*Corresponding author.

36th Conference on Neural Information Processing Systems (NeurIPS 2022).

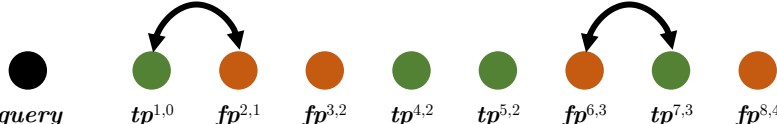

(a) True positives and their mis-ranks (next to ranks of true positives). the operation of swaps could happen between two side-by-side samples due to distances change.

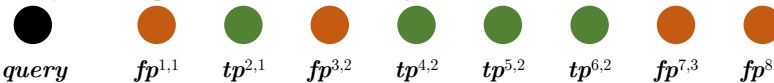

(b) After swapping, the involved true positives will have altered ranks and mis-ranks, while other true positives' will remain unchanged.

Figure 2: Mis-ranks marked on true positives and swaps that change ranks and mis-ranks.

**The side-by-side swaps.** Naturally, if a true positive and a false positive are placed side-by-side, and a swap happens between them due to the distances change, then mis-rank $m$ as well as rank $i$ of the true positive will be altered by 1 (Figs. 2(a) and 2(b)). Meanwhile, mis-ranks of any other true positives will remain unchanged. A special case is a side-by-side swap between two true positives or two false positives, where all ranks and mis-ranks would not change.

**Normal swaps.** More generally, any swaps happen in a rank list would influence ranks and mis-ranks of involved positive samples. To determine results after swaps, we could decompose the normal swaps into a series of side-by-side swaps with involved positive samples.

## B Proof

**Corollary.** Average precision increases *iff* $m$ decreases.

*Proof.* From the above demonstration, if any swap happens in a rank list between true and false positives, the mis-rank of that true positive is definitely changed and will only result in increase or decrease of $m$ and $i$ by 1. Correspondingly, precision at rank of involved true positive changes:

$$\frac{i \pm 1 - (m \pm 1)}{i \pm 1} = \frac{i - m}{i \pm 1},$$

which is reversely proportional to $m$. Since precision of other true positives' are unchanged, $AP$ is therefore influenced by the above true positive and reversely proportional to $m$. According to the derivation of normal swaps in Appendix A, this corollary will cover all cases of $AP$ calculations. $\square$

**Proposition.**

$$\overline{m} \propto \frac{\max d\left(\boldsymbol{q}, \boldsymbol{tp}\right)}{\min d\left(\boldsymbol{q}, \boldsymbol{fp}\right)} \ \forall \boldsymbol{tp} \in \boldsymbol{TP}, \ \boldsymbol{fp} \in \boldsymbol{FP}$$

where ‾ denotes upper bounds. Correspondingly,

$$\underline{AP} \propto \frac{\min d\left(\boldsymbol{q}, \boldsymbol{fp}\right)}{\max d\left(\boldsymbol{q}, \boldsymbol{tp}\right)} \ \forall \boldsymbol{tp} \in \boldsymbol{TP}, \ \boldsymbol{fp} \in \boldsymbol{FP}$$

where · denotes lower bounds.

*Proof.* Considering any true positive, we are able to derive following inequalities from it according to Eq. (1):

$$\min d\left(\boldsymbol{q}, \boldsymbol{fp}\right)_m < d\left(\boldsymbol{q}, \boldsymbol{tp}^m\right) < \min d\left(\boldsymbol{q}, \boldsymbol{fp}\right)_{m+1} \tag{2}$$

where $\min\left(\cdot\right)_m$ is the $m$-th minimum value among the whole set. For example, from Fig. 1, we could obtain:

$$\min d\left(\boldsymbol{q}, \boldsymbol{fp}\right)_3 < d\left(\boldsymbol{q}, \boldsymbol{tp}^{7,3}\right) < \min d\left(\boldsymbol{q}, \boldsymbol{fp}\right)_4 \tag{3}$$

where $\min d\left(\boldsymbol{q}, \boldsymbol{fp}\right)_3 = d\left(\boldsymbol{q}, \boldsymbol{fp}^6\right)$ and $\min d\left(\boldsymbol{q}, \boldsymbol{fp}\right)_4 = d\left(\boldsymbol{q}, \boldsymbol{fp}^8\right)$.

Then, for the last true positive in rank list, which has the largest $m$ among all true positives, *i.e.* $\overline{m}$, according to above inequalities, we could arrive:

$$\min d\left(\boldsymbol{q}, \boldsymbol{fp}\right)_{\overline{m}} < \max d\left(\boldsymbol{q}, \boldsymbol{tp}\right) < \min d\left(\boldsymbol{q}, \boldsymbol{fp}\right)_{\overline{m}+1}.$$

Now, if $\max d\left(\boldsymbol{q}, \boldsymbol{tp}\right)$ becomes smaller and $\min d\left(\boldsymbol{q}, \boldsymbol{fp}\right)_{\overline{m}}$ becomes larger so that a swap happens between the last true positive and its left most false positive, then we will have:

$$\overline{m}' = \overline{m} - 1, \ \ iff$$
$$\max d\left(\boldsymbol{q}, \boldsymbol{tp}\right)' < \max d\left(\boldsymbol{q}, \boldsymbol{tp}\right),$$
$$\min d\left(\boldsymbol{q}, \boldsymbol{tp}\right)'_{\overline{m}} > \min d\left(\boldsymbol{q}, \boldsymbol{tp}\right)_{\overline{m}} \ \ and$$
$$\min d\left(\boldsymbol{q}, \boldsymbol{tp}\right)'_{\overline{m}} > \max d\left(\boldsymbol{q}, \boldsymbol{tp}\right)'$$

where $(\cdot)'$ indicates the new value. Combining our assumption in Appendix A as well as the pigeonhole principle, increase of $\min d\left(\boldsymbol{q}, \boldsymbol{tp}\right)$ results in increase of $\min d\left(\boldsymbol{q}, \boldsymbol{tp}\right)_{\overline{m}}$. Therefore:

$$\left.\begin{array}{r}\max d\left(\boldsymbol{q}, \boldsymbol{tp}\right) \downarrow \\ \min d\left(\boldsymbol{q}, \boldsymbol{tp}\right) \uparrow \Rightarrow \min d\left(\boldsymbol{q}, \boldsymbol{tp}\right)_{\overline{m}} \uparrow\end{array}\right\} \overline{m} \downarrow,$$

and vice versa. This proves Eq. (2). Eq. (3) is immediately obtained since $AP$ is reversely proportional to $\overline{m}$ according to above corollary. □

**Proposition.**

$$\underline{AP} \propto \frac{\min d\left(\boldsymbol{q}, \boldsymbol{fp}\right)}{\max d\left(\boldsymbol{q}, \boldsymbol{tp}\right)} \geq const \cdot \frac{\min \mathcal{D}_{inter}}{\max \mathcal{D}_{intra}}. \tag{4}$$

*Proof.* Without loss of generality, $d\left(\boldsymbol{q}, \boldsymbol{tp}\right)$ is covered by distances between any two codes of the same class $c$ (denoted by $\mathcal{D}^c$): $d\left(\boldsymbol{q}, \boldsymbol{tp}\right) \subseteq \mathcal{D}^c = \left\{d\left(\boldsymbol{b}^i, \boldsymbol{b}^j\right), \ where \ y^i = y^j = c\right\}$. In contrast, $d\left(\boldsymbol{q}, \boldsymbol{fp}\right)$ is covered by: $d\left(\boldsymbol{q}, \boldsymbol{fp}\right) \subseteq \mathcal{D}^{\neq c} = \left\{d\left(\boldsymbol{b}^i, \boldsymbol{b}^j\right), \ where \ y^i = c, \ y^j \neq c\right\}$. For the first case, we have:

$$d\left(\boldsymbol{b}^i, \boldsymbol{c}^c\right) \leq \max \mathcal{D}_{intra}, \ d\left(\boldsymbol{b}^j, \boldsymbol{c}^c\right) \leq \max \mathcal{D}_{intra},$$
$$for \ any \ y^i = y^j = c$$

where $\boldsymbol{c}^c$ is the center of class $c$. Then, according to the triangle inequality:

$$d\left(\boldsymbol{b}^i, \boldsymbol{b}^j\right) \leq d\left(\boldsymbol{b}^i, \boldsymbol{c}^c\right) + d\left(\boldsymbol{b}^j, \boldsymbol{c}^c\right) \leq 2 \max \mathcal{D}_{intra}.$$

Therefore,

$$\max d\left(\boldsymbol{q}, \boldsymbol{tp}\right) \subseteq \mathcal{D}^c \leq 2 \max \mathcal{D}_{intra} = const \cdot \max \mathcal{D}_{intra}.$$

Similarly, $\min d\left(\boldsymbol{q}, \boldsymbol{fp}\right) \geq const \cdot \min \mathcal{D}_{inter}$ could be derived. Combined with two inequalities, the above proposition is proved. □

## B.1 Analysis on the Proposed Lower Bound

### B.1.1 Is the Introduced Lower Bound Tight?

To determine whether the lower bound is tight is a little bit difficult. We firstly introduce some concepts and assumptions to make it easier. Let us start at the example placed in beginning of Appendix A.

**Asm. 1.** Any positive samples do not have the same distances to query. This ensures Eq. (1).

**Asm. 2.** Noticed that we are working in the Hamming space where the Hamming distances between any two samples are discrete and range from 0 to $h$, Eq. (1) becomes:

$$0 \leq d\left(\boldsymbol{q}, \boldsymbol{tp}^1\right) < d\left(\boldsymbol{q}, \boldsymbol{fp}^2\right) < \cdots < d\left(\boldsymbol{q}, \boldsymbol{fp}^8\right) \leq h. \tag{5}$$

**Asm. 3.** The above array is strictly no gaps *i.e.* differences between any side-by-side $d\left(\boldsymbol{q}, \cdot\right)$ are 1.

Then, we would derive the closed form lowest $AP$ with $\frac{\min d(\boldsymbol{q},\boldsymbol{fp})}{\max d(\boldsymbol{q},\boldsymbol{tp})}$ under the same order of magnitude. This conclusion could be intuitively drawn from the above example where $\boldsymbol{tp}^{7,3}$ and $\boldsymbol{fp}^2$ determine $\max(\boldsymbol{q},\boldsymbol{tp}), \min(\boldsymbol{q},\boldsymbol{fp})$. If we keep two values unchanged (in other words, ranks of $\boldsymbol{tp}^{7,3}$ and $\boldsymbol{fp}^2$ unchanged), then the highest $AP$ will appear as the following figure shows:

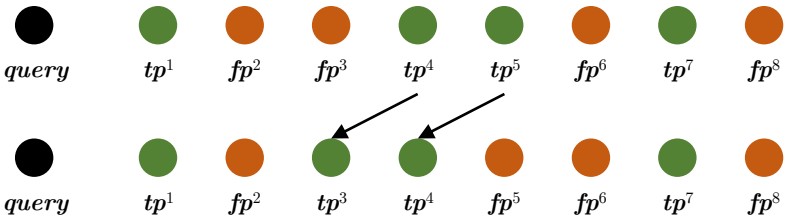

Figure 3: When $\max(\boldsymbol{q},\boldsymbol{tp}), \min(\boldsymbol{q},\boldsymbol{fp})$ are fixed, the highest $AP$ will appear when all true positives have higher ranks than false positives in-between $\max(\boldsymbol{q},\boldsymbol{tp}), \min(\boldsymbol{q},\boldsymbol{fp})$.

And the lowest $AP$ will appear as the following figure shows:

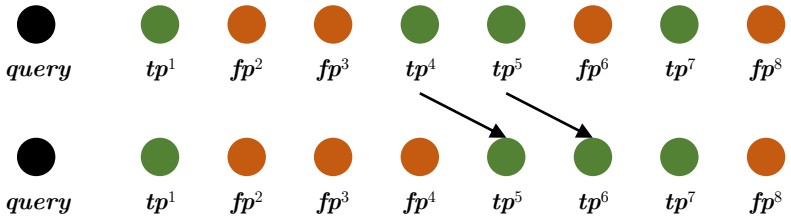

Figure 4: Correspondingly, the lowest $AP$ will appear when all true positives have lower ranks than false positives in-between $\max(\boldsymbol{q},\boldsymbol{tp}), \min(\boldsymbol{q},\boldsymbol{fp})$.

Based on this, we could easily derive that the lowest AP equals to

$$\min d(\boldsymbol{q},\boldsymbol{fp}) - 1 + \sum_{i=1}^{|\boldsymbol{TP}|} \frac{i}{\max d(\boldsymbol{q},\boldsymbol{tp}) - \min d(\boldsymbol{q},\boldsymbol{fp}) + i}$$

which is proportional to $\frac{\min d(\boldsymbol{q},\boldsymbol{fp})}{\max d(\boldsymbol{q},\boldsymbol{tp})}$. Therefore, the lower bound is tight.

We could further extend $\frac{\min d(\boldsymbol{q},\boldsymbol{fp})}{\max d(\boldsymbol{q},\boldsymbol{tp})}$ to $\frac{\min \mathcal{D}_{inter}}{\max \mathcal{D}_{intra}}$ for the tight lower bound. From Eq. (4), we find that $\frac{\min d(\mathbf{q},\mathbf{fp})}{\max d(\mathbf{q},\mathbf{tp})} \geq \frac{\min \mathcal{D}_{inter}}{\max \mathcal{D}_{intra}}$. The equality is achieved when query's code $\mathbf{q}$ is exactly the same as its class-center $\mathbf{c}$'s code. In this condition, the tight lower bound is derived by $\frac{\min \mathcal{D}_{inter}}{\max \mathcal{D}_{intra}}$.

Then, could it be applied to general cases? We give our humble discussion for a simple study. There may have untouched complicated cases that will be leaved for future study.

**Case 1: Duplicated positives.** If some samples are hashed to the same binary code (collision), then distances from query to them are all equal. They will appear at the same position of the rank list. If they are all true positives or false positives, then we could directly treat them as a single duplicated sample and follow the above rules. For example:

The above rank list has the duplicated true positives ($d(\boldsymbol{q}, \boldsymbol{tp}^{4,2}) = d(\boldsymbol{q}, \boldsymbol{tp}^{5,2})$). If a swap happens between them and $\boldsymbol{fp}^6$, the rank list will become:

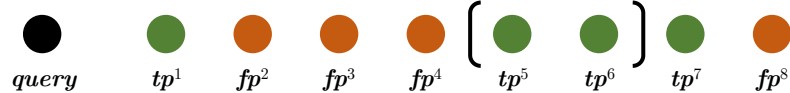

where $i, m$ of the duplicated true positives are both increased by 1. Obviously, the lower bound is still tight.

**Case 2: Mixed positives.** It is tricky when true positives and false positives have the same distance with query (we call them mixed positives). The sorting algorithm to produce final rank list also has impact to determine ranks of these mixed positives. Whether the lower bound is tight in this case is hard to judge, but our lower bound is still valid since above corollary still makes sense.

**Case 3: Rank list with gaps.** If there are gaps in between two distances, *e.g.*, $d\left(\boldsymbol{q}, \boldsymbol{fp}^i\right) \ll d\left(\boldsymbol{q}, \boldsymbol{tp}^{i+1}\right)$ (this usually happens on outliers), then $AP$ will be far from the lower bound. Please refer to Appendix B.2 for details.

In conclusion, under the above assumptions Asm. 1, 2, 3, our lower bound is tight. Meanwhile, our lower bound also covers common cases as in the above discussion and makes a strong connection to $AP$.

### B.2 Edge Cases of the Lower Bound

Some edge cases when the lower bound is far way from the $AP$ are further revealed according to the above analysis.

**Edge Case 1.** A huge amount of samples are hashed to the same binary code. Based on Case 1 and 2 in Appendix B.1.1, we will observe many duplicated or mixed samples. Their ranks will be increased / decreased simultaneously, and AP significantly changes along with them. The rank list is now be treated as "unstable". Intuitively, when a hash-model has poor performance, it could not distinguish differences between samples and simply hashes them to the same code, and such a model will produce "unstable" rank lists.

**Edge Case 2.** Gaps in rank list. In the above example, if $d\left(\boldsymbol{q}, \boldsymbol{fp}^6\right) \ll d\left(\boldsymbol{q}, \boldsymbol{tp}^{7,3}\right) = \max d\left(\boldsymbol{q}, \boldsymbol{TP}\right)$, $\max d\left(\boldsymbol{q}, \boldsymbol{TP}\right)$ needs to be significantly decreased until a swap happens between $\boldsymbol{fp}^6$ and $\boldsymbol{tp}^{7,3}$ to influence $AP$. Therefore, $AP$ is high but $\frac{\min\left(\boldsymbol{q}, \boldsymbol{fp}\right)}{\max\left(\boldsymbol{q}, \boldsymbol{tp}\right)}$ is low. A potential reason leads to this edge case is outliers.

## C Criterion of Hash Codes' Performance

Hashing techniques are widely applied in common multimedia tasks. To determine hash codes' performance, the direct way is to adopt evaluation metrics used in these tasks *e.g.* retrieval precision, recall or recognition accuracy. In this section, we give the detailed explanation on how to adopt $AP$ to cover the above typical metrics.

**Retrieval.** Common evaluation metrics in retrieval can be derived from $AP$. Specifically,

- Precision at rank $i$ equals to $\frac{i-m}{i}$. The corollary in Appendix B exactly applies to it.

- Recall at rank $i$ equals to $\frac{i-m}{|\boldsymbol{T}|}$ where $\boldsymbol{T}$ is set of all groundtruth samples. Therefore, $|\boldsymbol{T}|$ is a constant and recall increases *iff* $m$ decreases.

- F-score equals to $\frac{2}{recall^{-1}+precision^{-1}} = \frac{2}{i/(i-m)+|\mathbf{T}|/(i-m)} = \frac{2(i-m)}{i+|\mathbf{T}|}$.

All the above metrics are reversely proportional to $m$. Therefore, analysis in Appendix B is also valid to them.

**Recognition.** There are many ways to adopt hash codes for recognition, and we discuss two main approaches here. 1) $k$NN based recognition is very similar to retrieval. In this scenario, prediction is obtained by voting among top-$k$ retrieved results. If there are more than half true positives in rank list, then the prediction will be correct, otherwise it will be wrong. Therefore, the recognition accuracy can be treated as a special case where we hold a rank list of all samples and expect $m < k/2$ when $i = k$, which is induced to the goal of increasing $AP$.

As for logistic regression models for recognition, a learnable weight is adopted to make predictions: $\boldsymbol{b} \xrightarrow{\boldsymbol{w}} \boldsymbol{cls}$, $\boldsymbol{w} \in \mathbb{R}^{C \times h}$ where the output $\boldsymbol{cls}$ is $C$-dim scores for each class. We could treat $\boldsymbol{w}$ as $C$ class-prototypes. Each prototype $\boldsymbol{w}^i$ is a $h$-dim vector belongs to class $i$ while $\boldsymbol{cls}_i$ measures inner-product similarity between prototype and hash code $\boldsymbol{b}$. Inner-product similarity is an approximation of Hamming distance and $\boldsymbol{w}$ is trained by cross-entropy objective, resulting in maximizing $\boldsymbol{cls}_c$ while minimizing $\boldsymbol{cls}_{other}$. Such a goal is equivalent to maximizing intra-class compactness and inter-class distinctiveness.

### C.1 Potential Use Cases of the Lower Bound

Exploring use cases of the proposed lower bound would reveal significance and value of this work. Since Figure 1 and 5 in main paper and the above analysis tell us the value of $\frac{\min \mathcal{D}_{inter}}{\max \mathcal{D}_{intra}}$ partially reflects hash codes' performance, we would quickly evaluate model's performance by such a metric other than calculating $AP$ or accuracy which is time consuming. Therefore, adopting it as a performance indicator benefits for model tuning or selection, including but not limited to parameter search.

## D  Multivariate Bernoulli Distribution

In this paper, we treat the posterior of hash codes $p(\mathcal{B} \mid \mathcal{X})$ is under Multivariate Bernoulli distribution $\mathrm{MVB}$. This is based on our observations of hash-model outputs. To demonstrate this claim, a toy 2 bits CSQ model is trained on 2 randomly chosen classes of CIFAR-10. We then plot $\boldsymbol{\ell}$ extracted by CSQ in Fig. 5.

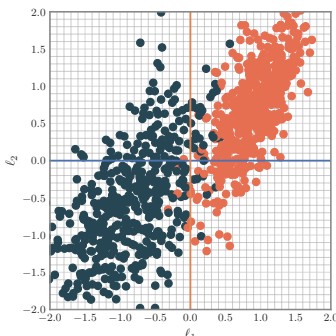

Figure 5: $\boldsymbol{\ell}$ extracted from a toy 2 bits CSQ model with 2 classes' samples of CIFAR-10. Correlations exist in $\boldsymbol{b}_1$, $\boldsymbol{b}_2$.

|         | $p(\boldsymbol{b}_1)$ | $\boldsymbol{b}_1 < 0$ | $\boldsymbol{b}_1 > 0$ |
|---|---|---|---|
| $p(\boldsymbol{b}_2)$ |  | 0.473 | 0.527 |
| $\boldsymbol{b}_2 > 0$ | 0.525 | 0.079 | 0.446 |
| $\boldsymbol{b}_2 < 0$ | 0.475 | 0.394 | 0.081 |

Figure 6: Estimated joint and marginal probabilities. Obviously, joint probability is not equal to product of marginal probability.

Obviously, features are concentrated on the upper-right and lower-left regions on the figure. This indicates that hash codes of these samples are tend to be (+1+1) or (-1-1) while very few samples are hashed to (+1-1) or (-1+1). Based on this pattern, we could infer that if $\boldsymbol{b}_1$ and $\boldsymbol{b}_2$ have a positive correlation. We also count frequencies of all 4 possible code combinations and calculate the estimated joint and marginal probability in Fig. 6, which confirms our observation. As the figure and the table show, joint probabilities are not equal to product of marginal probabilities, meaning that correlation exists among codes. Therefore, $\mathrm{MVB}$ is suitable to model the posterior of hash codes. Previous works are not sufficient to model it since they process each hash bit independently.

For instance, if $h = 4$, joint probability of $\boldsymbol{b} = (+1+1\text{-}1\text{-}1)$ is formulated as:

$$p(\boldsymbol{b} = (+1+1\text{-}1\text{-}1)) = p(\boldsymbol{b}_1 < 0, \boldsymbol{b}_2 < 0, \boldsymbol{b}_3 > 0, \boldsymbol{b}_4 > 0). \tag{6}$$

Unfortunately, estimating this joint probability is difficult [2, 3, 6]. Furthermore, all $2^h$ joint probabilities are required to be estimated in order to control hash codes precisely. Therefore, we propose our posterior estimation approach to tackle above difficulties.

## E  Demonstration of Posterior Estimation

We continue to use the example in Eq. (6) to explain how we build our surrogate model to perform posterior estimation. Firstly, 4 bits hash codes have all 16 choices from (-1-1-1-1) to (+1+1+1+1). Therefore, $o$ has 16 entries where each entry in $o$ will be used to estimate a specific joint probability. We provide a simple demonstration in Fig. 7 to show how we process it.

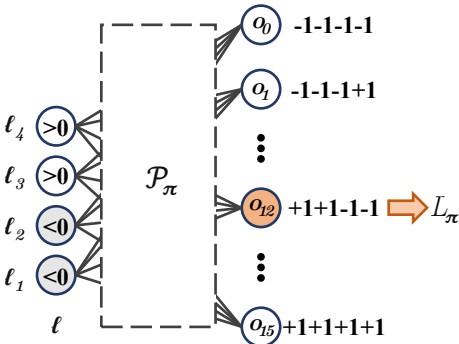

Figure 7: Demonstration of how $\mathcal{P}_{\pi}$ performs posterior estimation.

Specifically, if we feed $\ell$ where $\ell_1 < 0$, $\ell_2 < 0$, $\ell_3 > 0$, $\ell_4 > 0$ *i.e.* $b = (+1+1\text{-}1\text{-}1)$, then we could directly find output $o_{12} \triangleq p\,(b_1 < 0, b_2 < 0, b_3 > 0, b_4 > 0)$ where $12 = 1 \cdot 2^3 + 1 \cdot 2^2 + 0 \cdot 2^1 + 0 \cdot 2^0$ and perform MLE on it to train model $\pi$.

### E.1  Implementation of Posterior Estimation

Correspondingly, we provide a minimal PyTorch-style implementation of $\mathcal{P}_{\pi}$ and its training objective in Fig. 8. We first give one of a implementation of non-linear $\ell$ to $o$ mapping in line № 1~8. Then, the main body of surrogate model (line № 11~55) shows how we process codes in blocks (line № 49) and how we convert every 8 bits inputs to 256 categorical outputs and calculate loss *w.r.t.* targets (line № 53).

## F  Additional Experiments

We continue to conduct experiments for more comparisons and ablations. Before this, we firstly provide more implementation details as a supplement to main paper.

As Fig. 8 shows, we choose a normal two layer model with SiLU activation as $\mathcal{P}_{\pi}$ to perform posterior estimation. To avoid over-fitting, we insert a dropout layer with 0.5 probability between layers. We train all methods for 100 epoches with batch-size 64, and learning rate is exponential decayed by 0.1 every 10 epoches. Most of the methods could achieve 95% performance after 30 epoches. We do not perform grid-search on hyper-parameters to obtain the highest results since it is not our topic in this paper.

To generate centers of specific classes for training (Alg.1 Line.2), we want a *perfect* generator which could produce finite number of codes with maximized pairwise Hamming distance among each other. Such a problem is formulated as the $A_q\,(n, d)$ problem [4, 5] and such perfect codes are recognized to attain the *Hamming bound* [1, 7, 8]. However, this is extremely hard to tackle. Therefore, we conduct ablation study on a series of error-correction codes which aims at approaching the above goal to pick the best one, which will be explained below.

```python
1   # One of the implementations for non-linear
2   # mapping to be used in surrogate model.
3   _nonLinearNet = lambda: nn.Sequential(
4       nn.Linear(8, 256),
5       nn.SiLU(),
6       nn.Dropout(0.5),
7       nn.Linear(256, 256)
8   )
9
10
11  class Surrogate(nn.Module):
12      """
13      Surrogate model to estimate MVB and further provide gradients.
14
15      Args:
16          bits (int): Length of hash codes.
17      """
18      def __init__(self, bits: int):
19          super().__init__()
20          # Number of blocks.
21          self._u = bits // 8
22          self._bits = bits
23          self._net = nn.ModuleList(
24              _nonLinearNet() for _ in range(self._u)
25          )
26          # binary to decimal multiplier
27          self.register_buffer("_multiplier",
28              (2 ** torch.arange(8)).long())
29          self._initParameters()
30
31      def forward(self, x: Tensor, t: Tensor = None) -> (Tensor):
32          """
33          Module forward.
34          Args:
35              x (Tensor): [N, h] Model outputs before `.sign()`
36                  (un-hashed values).
37              t (Tensor, Optional): [N, h] Target hash result of x.
38          Return:
39              Tensor: [] If t is None, produces loss to
40                  train surrogate model. Otherwise,
41                  produces loss to calculate surrogate
42                  gradient w.r.t. x.
43          """
44          loss = list()
45          t = t or x
46          # split hash codes into `u` pieces, each 8 bits.
47          for subNet, org, tgt in zip(self._net,
48              torch.chunk(x, self._u, -1),
49              torch.chunk(t, self._u, -1)):
50              # feed each 8 bits codes into i-th non-linear model
51              # to get [256] predictions.
52              prd = subNet(org)
53              y = ((tgt > 0) * self._multiplier).sum(-1)
54              loss.append(F.cross_entropy(prd, y))
55          return sum(loss)
```

Figure 8: Minimal implementation of our proposed surrogate model for posterior estimation.

## F.1 Performance on Large Dataset

To evaluate performance of our proposed method on large-scale data, we conduct a new experiment on the whole ImageNet (ILSVRC 2012). Specifically, this dataset includes $1,000$ classes with a relatively large training set ($1.2M$ images, ~$10\times$ larger than datasets adopted in main paper). The

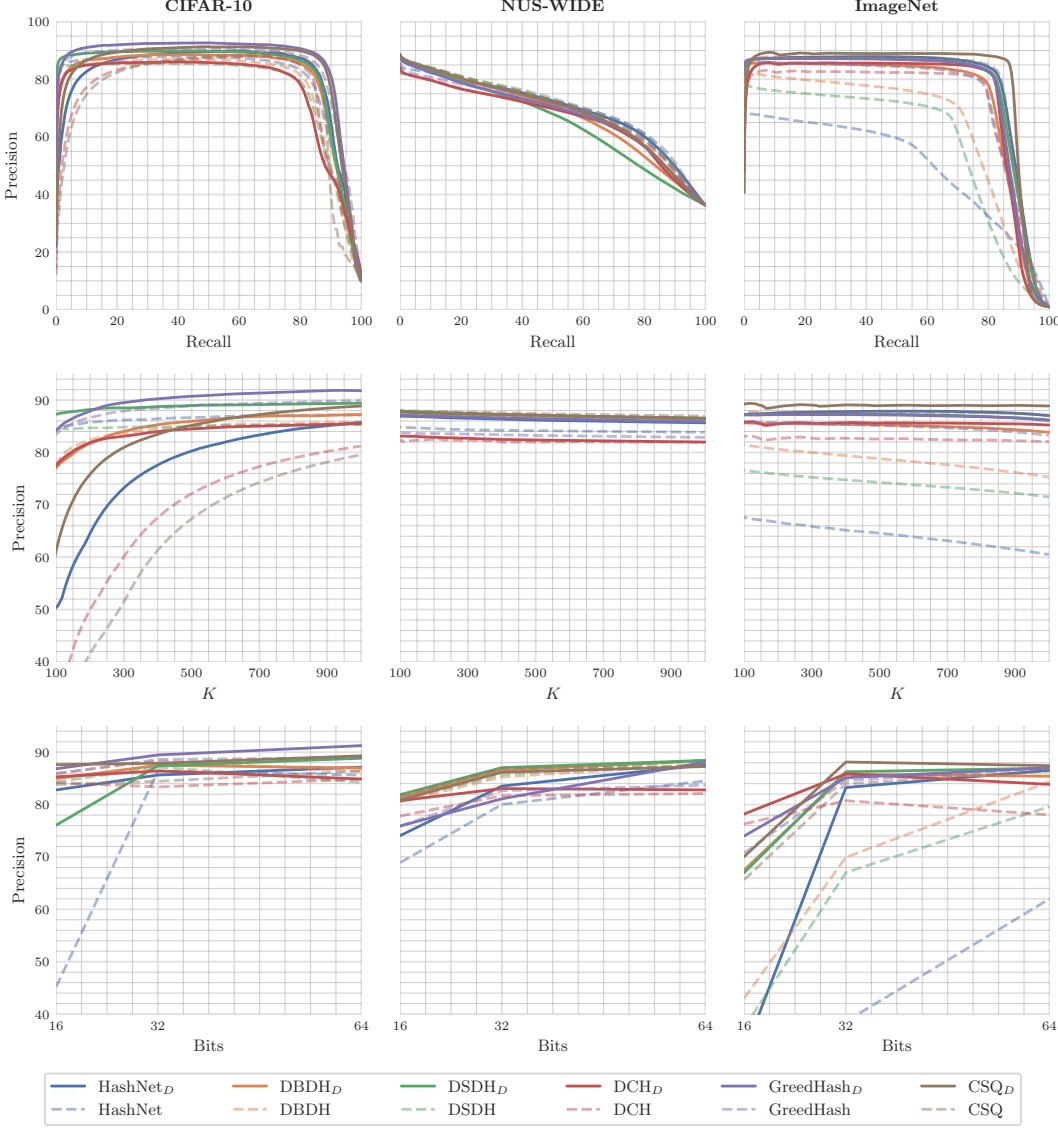

Figure 9: **First row**: Precision-Recall (P-R) curves for three datasets on $64$ bits. With our method integrated, most of methods' P-R curves slightly move to upper right.
**Second row**: Precision@$K$ (P@$K$, $K$ from $100$ to $1,000$) curves for three datasets on $64$ bits. With our method integrated, most of methods' P@$K$ curves slightly go up.
**Third row**: Precision@$H = 2$ (retrieval inside the $H = 2$ Hamming ball) *w.r.t.* code-length curves on three datasets.

validation set has $50,000$ images and $50$ for each class for retrieval. We randomly pick $5$ images of each class as queries ($5,000$ in total) and the remaining is adopted to formulate the base split ($45,000$ in total). Networks are trained for $20$ epoch and we only update the last hash layers since the backbone is already pre-trained on ImageNet. Other settings are the same with main paper. From Tab. 1, we could see our method is also effective when trained with large dataset.

## F.2 Quantitative Comparisons

**Precision-Recall Curves.** To indicate retrieval performance over the whole rank list, we plot precision-recall curves for all methods in the first row of Fig. 9 for three datasets on $64$ bits. We obtain similar results with Tab.2 where all the methods obtain performance gain (curves move to upper-right) after integrating our methods.

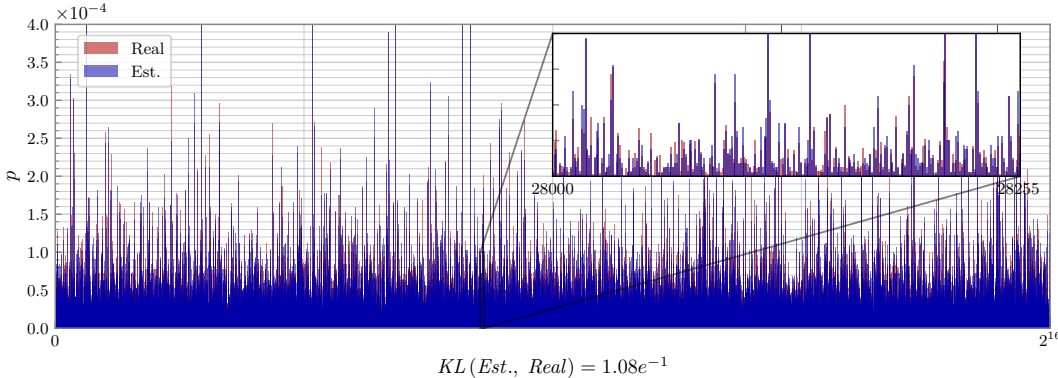

$$KL\left(Est.,\ Real\right)=1.08e^{-1}$$

Figure 10: Posterior estimation on 16 bits codes ($65,536$ joint probabilities) with a block=2 surrogate model. We randomly show a zoomed-in view with interval of 256 on the upper-right. Ours $KL$: 0.1084, naïve's $KL$: 0.4965.

**Precision@$K$ Curves.** To visualize retrieval precision of top retrieved samples, Precision@$K$ ($K$ from $100$ to $1,000$) curves for three datasets on $64$ bits are placed in the second row of Fig. 9. Similar to P-R curves, most of the methods receive a precision increase and curves slightly move up.

**Precision@$H = 2$ Curves.** To show retrieval performance inside $H = 2$ Hamming ball, we draw Precision@$H = 2$ *w.r.t.* code-length curves on the third row of Fig. 9. With our methods, precision inside Hamming ball is also increased. This indicates that our method pushes more true positives close to queries than the original methods.

### F.3 Ablation Study

#### F.3.1 Posterior Estimation on Longer Bits.

Due to space limitation in main paper, we place explanation on how we estimate MVB by naïve way here (Sec.6.2). Then, an extra experiments on longer bits *i.e.* 16 bits will be conducted to validate the effectiveness of blocked estimation.

For naïve estimation on MVB, dependence among $\boldsymbol{b}$ is ignored. Specifically, we directly count frequency of value to be greater than zero on each bit of codes to approximate $p\left(\boldsymbol{b}_i > 0\right)$ and $p\left(\boldsymbol{b}_i < 0\right)$. Then, any joint probability is estimated by the product of marginal probabilities, *e.g.* $p\left(\boldsymbol{b} = +1+1\text{-}1\text{-}1\right) \;\widehat{=}\; p\left(\boldsymbol{b}_4 > 0\right) \cdot p\left(\boldsymbol{b}_3 > 0\right) \cdot p\left(\boldsymbol{b}_2 < 0\right) \cdot p\left(\boldsymbol{b}_3 < 0\right)$.

Table 1: Retrieval performance on the full ImageNet dataset.

| Method | 16 bits | 32 bits | 64 bits |
|---|---|---|---|
| HashNet | 36.9 | 41.2 | 44.7 |
| HashNet$_D$ | **55.0**$_{\uparrow 18.1}$ | **56.1**$_{\uparrow 14.9}$ | **61.8**$_{\uparrow 17.1}$ |
| DBDH | 37.4 | 41.1 | 49.1 |
| DBDH$_D$ | **57.2**$_{\uparrow 19.8}$ | **57.9**$_{\uparrow 16.8}$ | **60.4**$_{\uparrow 11.3}$ |
| DSDH | 39.3 | 48.4 | 54.2 |
| DSDH$_D$ | **56.0**$_{\uparrow 16.7}$ | **59.1**$_{\uparrow 10.7}$ | **62.0**$_{\uparrow 7.8}$ |
| DCH | 60.2 | 62.3 | 64.1 |
| DCH$_D$ | **61.9**$_{\uparrow 1.7}$ | **62.9**$_{\uparrow 0.6}$ | **64.4**$_{\uparrow 0.3}$ |
| GreedHash | 62.7 | 63.1 | 64.9 |
| GreedHash$_D$ | **63.1**$_{\uparrow 0.4}$ | **64.0**$_{\uparrow 0.9}$ | **65.9**$_{\uparrow 1.0}$ |
| CSQ | 64.6 | 65.0 | 65.7 |
| CSQ$_D$ | **65.7**$_{\uparrow 1.1}$ | **66.2**$_{\uparrow 1.2}$ | **66.4**$_{\uparrow 0.7}$ |

To train on 16 bits codes, we adopt a block=2 surrogate model where the first block is used to produce the first $8$ bits codes and the second one is for the last $8$ bits (as in Fig. 8 implements). Then, for all $65,536$ joint probabilities, we obtain them by the Cartesian product of two models' predictions, which is shown in Fig. 10. The figure shows similar results as the $8$ bits experiment in main paper. Our model achieves a much lower $KL$ than naïve one, showing that performing block estimation will not introduce significant bias to estimate posterior (Ours: 0.1084, naïve: 0.4965).

Table 2: mAP comparisons with different pre-defined centers on three benchmark datasets for $16, 32, 48$ bits codes. Values on lower right is minimum pairwise Hamming distances among centers.

| Center | CIFAR-10 | | | NUS-WIDE | | | ImageNet | | |
|---|---|---|---|---|---|---|---|---|---|
| | 16 bits | 32 bits | 48 bits | 16 bits | 32 bits | 48 bits | 16 bits | 32 bits | 48 bits |
| Reed-Solomon | $87.7_6$ | $89.0_{14}$ | $89.8_{21}$ | $83.1_5$ | $84.9_{12}$ | $85.0_{19}$ | $88.5_4$ | $89.5_{10}$ | $90.2_{16}$ |
| BCH | $88.3_7$ | $89.2_{16}$ | $89.9_{21}$ | $83.0_4$ | $84.8_{12}$ | $85.1_{19}$ | $87.3_2$ | $89.0_9$ | $89.8_{14}$ |
| Hadamard | $88.7_8$ | $89.2_{16}$ | $89.5_{19}$ | $83.3_8$ | $85.3_{16}$ | $85.0_{19}$ | $87.9_3$ | $89.0_8$ | $89.8_{14}$ |

### F.3.2  Impact of Different Kinds of Pre-defined Centers.

To generate pre-defined centers as separate as possible as targets to train model, we compare a series of error-correction codes. The experiment includes Reed-Solomon code, BCH code and Hadamard code. Four kinds of codes have different pairwise Hamming distance. According to our main proposition in paper, if the minimum pairwise Hamming distance between centers are smaller, the lower bound of hash codes' performance is tend to be correspondingly lower. To show this phenomenon, minimum pairwise Hamming distance of codes as well as mAP are measured for $CSQ_D$, which is placed in Tab. 2. It is worth noting that Hadamard code is not applicable when number of centers is large or code-length is not order of 2, in this case we report results with randomly generated codes. The table confirms the positive correlation between minimum pairwise Hamming distance and performance. Meanwhile, performance differences among three kinds of codes are small. Therefore in practice, we could choose any of them.