# OpenReview forum: "A Lower Bound of Hash Codes' Performance"
_NeurIPS.cc/2022/Conference — NeurIPS 2022 Accept_

### Official Review · Reviewer_jH1U · 2022-07-10

**Rating:** 6
**Confidence:** 3
**Soundness:** 3 good
**Presentation:** 3 good
**Contribution:** 3 good

**Summary:**

This paper studies the learn-to-hash problem, where we need to transform images into hash codes for fast retrieval.  The major research highlight of this paper is the lower bound for hash codes' performance. This paper uses inter-class distinctiveness and intra-class compactness to present a lower bound for the average precision, an evaluation metric for learn-to-hash. Moreover, this paper uses this lower bound as an objective for learn-to-hash. As a result, this paper presents a significant increase in average precision and accuracy.

**Questions:**

1. This paper introduces a lower bound for AP using inter-class distinctiveness and intra-class compactness. Is this lower bound tight? On which situation the lower bound is equal to the AP? Is there any intuition on the edge cases that the lower bound is far way from the AP?

2. This paper introduces the strength of the proposed lower bound as an objective function. Is there any more potential use case for this lower bound other than the objective? For instance, can we use this lower bound in the parameter search of hash code length?

3. This paper presents improvements in both mAP and accuracy using the proposed lower bound as an objective. Since the lower bound is designed for AP. Is there a connection between it and the accuracy so that we observe this improvement. In fact, this leads to a more general question: how does this lower bound connect to other metrics such as recall, and precision?


**Limitations:**

In general, this paper has a nice presentation and organization. Both theoretical analysis and experimental evaluation are presented for better illustration. However, it would be better to provide a deeper analysis of the functionality of the lower bound. I would like to see the answers to the raised questions in the previous section.

**Strengths And Weaknesses:**

Strengths;

1. The proposed lower bound is useful. It connects the evaluation metric with the current state of the learn-to-hash model. As a result, we can use it as a guide in the model training.

2. The experimental evaluation is extensive. The authors present a comparison with solid baselines. Moreover, the authors study the training efficiency of the proposed objective with Cauchy and BCE.

3.  The paper provides a theoretical analysis of the proposed lower-bound and its gradient when we use it as a loss. In the supplementary material, this paper also includes necessary implementation details.

---

> ### Author Response · Authors · 2022-08-02
> **[2/2] Response to Reviewer jH1U**
>
>
> [**Part 2 of 2**]
>
> ---
>
> ### ***Q2: On which situation the lower bound is equal to the AP?***
> Under above assumptions in `Q1`, the lowest AP has a closed form derived by our proposed lower bound $\frac{\min{\left(\mathbf{q}, \mathbf{fp}\right)}}{\max{\left(\mathbf{q}, \mathbf{tp}\right)}}$ under the same order of magnitude.
>
> Also from the example:
> $$
> \mathbf{query}  \Rightarrow  \mathbf{tp}^{1,0} ; \mathbf{fp}^2 ; \mathbf{fp}^3 ; \mathbf{tp}^{4,2} ; \mathbf{tp}^{5,2} ; \mathbf{fp}^6 ;  \mathbf{tp}^{7,3} ; \mathbf{fp}^8
> $$
> where $\mathbf{tp}^{7,3}$ and $\mathbf{fp}^2$ determine $\max{\left(\mathbf{q}, \mathbf{tp}\right)}, \min{\left(\mathbf{q}, \mathbf{fp}\right)}$. If we keep two values unchanged (in other words, ranks of two samples unchanged), then, the highest AP will appear when:
> $$
> \mathbf{query}  \Rightarrow  \mathbf{tp}^{1,0} ; \mathbf{fp}^2 ;  \mathbf{tp}^{3,1} ; \mathbf{tp}^{4,1} ; \mathbf{fp}^5 ; \mathbf{fp}^6 ; \mathbf{tp}^{7,3} ; \mathbf{fp}^8.
> $$
> And the lowest AP will appear when:
> $$
> \mathbf{query}  \Rightarrow  \mathbf{tp}^{1,0} ; \mathbf{fp}^2 ; \mathbf{fp}^3 ; \mathbf{fp}^4 ;  \mathbf{tp}^{5,3}  \mathbf{tp}^{6,3} ; \mathbf{tp}^{7,3} ; \mathbf{fp}^8.
> $$
>
> Based on this, we could easily derive that the lowest AP equals to
> $$
> \min{d\left(\mathbf{q}, \mathbf{fp}\right)} - 1 + \sum_{i=1}^{\lvert \mathbf{TP} \rvert}{\frac{i}{\max{d\left(\mathbf{q}, \mathbf{tp}\right)} - \min{d\left(\mathbf{q}, \mathbf{fp}\right)} + i}}
> $$
> which is proportional to $\frac{\min{\left(\mathbf{q}, \mathbf{fp}\right)}}{\max{\left(\mathbf{q}, \mathbf{tp}\right)}}$.
>
> ---
>
> ### ***Q3: Is there any intuition on the edge cases that the lower bound is far way from the AP?***
> According to answer of `Q1`, some edge cases are further revealed.
>
> * **Edge case 1**: A huge amount of samples are hashed to the same binary code.\
> Based on `case 1 and 2 in Q1`, here, we will observe many duplicated or mixed samples. Their ranks will be increased / decreased simultaneously, and AP will be significantly changed along with them. We think now the rank list is "unstable". Intuitively, this unstability is caused by the poor hashing model, since it could not distinguish differences between samples and simply hashes them to the same code.
> * **Edge case 2**: Gaps in rank list.\
> For example, if:
> $$
> \mathbf{query}  \Rightarrow  \mathbf{tp}^{1,0} ; \mathbf{fp}^2 ; \mathbf{fp}^3 ; \mathbf{tp}^{4,2}  \mathbf{tp}^{5,2} ; \mathbf{fp}^6 ;  \mathbf{tp}^{7,3} ; \mathbf{fp}^8,
> $$
> where $d\left(\mathbf{q}, \mathbf{fp}^6\right) \ll d\left(\mathbf{q}, \mathbf{tp}^{7, 3}\right) = \max{d\left(\mathbf{q}, \mathbf{TP}\right)}$, then, to influence AP, $\max{d\left(\mathbf{q}, \mathbf{TP}\right)}$ needs to be significantly decreased until a swap happens between $\mathbf{fp}^6$ and $\mathbf{tp}^{7, 3}$. Therefore, AP may be still high but $\frac{\min{\left(\mathbf{q}, \mathbf{fp}\right)}}{\max{\left(\mathbf{q}, \mathbf{tp}\right)}}$ is low. We think outliers cause this edge case.
>
> ---
>
>
> ### ***Q4: Potential use cases of lower bound.***
> **A:** Exploring use cases of proposed lower bound would reveal significance and value of this work. Just as you mentioned in review, such lower bound could be adopted as a criterion for *e.g.* parameter search of hash code length. Since `Figure 1, 5` in paper and above analysis tell us the value of $\frac{\min{\mathcal{D}_\mathit{inter}}}{\max{\mathcal{D}_\mathit{intra}}}$ partially reflects hash model's performance, we would quickly evaluate model's performance by such metric other than calculating AP or accuracy which is time consuming. Therefore, adopting it as a performance indicator benefits for model tuning or selection, including but not limited to parameter search.
>
> ---
>
>
> ### ***Q5: How does this lower bound connect to other metrics?***
> **A:** From `Q1`'s answer, we already know the proposed lower bound has a strong connection between AP. As for other metrics including *precision*, *recall*, *F-score*, *accuracy*, *etc.*, we have discussed how AP is related to them in `Supp. Sec. C.`. Specifically,
>
> * Precision at rank $i$ equals to $\frac{i - m}{i}$. The corollary in `Supp. Sec. B` exactly applies to it.
> * Recall at rank $i$ equals to $\frac{i - m}{\lvert \mathbf{T} \rvert}$ where $\mathbf{T}$ is set of all groundtruth samples. Therefore, $\lvert \mathbf{T} \rvert$ is a constant and recall increases $\mathit{iff}$ &nbsp; $m$ decreases.
> * F-score equals to $\frac{2}{\mathit{recall}^{-1} + \mathit{precision}^{-1}} = \frac{2}{i / \left(i - m\right) + \lvert \mathbf{T} \rvert / \left(i - m\right)} = \frac{2\left(i - m\right)}{i + \lvert \mathbf{T} \rvert}$.
>
> We could see all above metrics are reversely proportional to $m$. Then, analysis in `Supp. Sec. B` is also valid to them.
>
> As for accuracy, please refer to `Supp. Sec. C.` for detailed explanation.
>
> ---
>
> \
> Thanks again for your kind review and valuable suggestions. All above analysis is in the rebuttal revision of supplementary materials.
>
> \
> \
> Best,
>
> Paper 44 Authors

---

> ### Author Response · Authors · 2022-08-02
> **[1/2] Response to Reviewer jH1U**
>
>
> [**Part 1 of 2**]
>
> ---
>
> Thanks for your review and valuable suggestions for us to improve our work. Next we would address your concerns with detailed explanations.
>
> ### ***Q1: Is the introduced lower bound tight?***
> **A:** It is a very interesting questions, and we hope following analysis could explain it based on characteristics of our proposed lower bound.
>
> To determine whether the lower bound is tight is a little bit difficult. We first introduce some concepts and assumptions to make it easier. Let us start at the example placed in beginning of `Supp. Sec. A`.
>
> * **Asm. 1**. Just as mentioned in Supp., we assume that any positive samples do not have the same distances to query. This ensures `Eq. (1)` in `Supp. Sec. A`.
> * **Asm. 2**. Noticed that we are working in the Hamming space where Hamming distances between any two codes are discrete and range from $0$ to $h$, `Eq. (1)` becomes:
> $$
> 0 \leq d\left(\mathbf{q}, \mathbf{tp}^1\right) < d\left(\mathbf{q}, \mathbf{fp}^2\right) < \cdots < d\left(\mathbf{q}, \mathbf{fp}^8\right) \leq h.
> $$
> * **Asm. 3**. Let above array be strictly without gaps *i.e.* differences between any side-by-side $d\left(\mathbf{q}, \cdot\right)$ are $1$.
>
> Then, we would derive the closed form lowest AP by $\max{d\left(\mathbf{q}, \mathbf{tp}\right)}$ and $\min{d\left(\mathbf{q}, \mathbf{fp}\right)}$ (detailed in `Q2`), which is proportional to $\frac{\min{d\left(\mathbf{q}, \mathbf{fp}\right)}}{\max{d\left(\mathbf{q}, \mathbf{tp}\right)}}$.
>
> Since the lowest AP is derived by $\frac{\min{d\left(\mathbf{q}, \mathbf{fp}\right)}}{\max{d\left(\mathbf{q}, \mathbf{tp}\right)}}$ and under the same order of magnitude, we say the proposed lower bound is tight.
>
> We could further expand $\frac{\min{d\left(\mathbf{q}, \mathbf{fp}\right)}}{\max{d\left(\mathbf{q}, \mathbf{tp}\right)}}$ to $\frac{\min{\mathcal{D}_\mathit{inter}}}{\max{\mathcal{D}_\mathit{intra}}}$ for tight lower bound. From the proposition in main paper, we find that $\frac{\min{d\left(\mathbf{q}, \mathbf{fp}\right)}}{\max{d\left(\mathbf{q}, \mathbf{tp}\right)}} \geq \frac{\min{\mathcal{D}_\mathit{inter}}}{\max{\mathcal{D}_\mathit{intra}}}$. The equality is achieved when query's code $\mathbf{q}$ is exactly the same as its class-center's code $\mathbf{c}$. In this circumstance, the tight lower bound is derived by $\frac{\min{\mathcal{D}_\mathit{inter}}}{\max{\mathcal{D}_\mathit{intra}}}$.
>
> Now, could it be applied to general cases? We give our humble discussion for a simple study. There may have untouched complicated cases which are leaved for future study.
>
> * **Case 1**: Duplicated positives.\
> If some samples are hashed to the same binary code, then distances from query to them are all equal. They will appear at the same position of rank list. If they are all true positives or false positives, then we could directly treat them as a single (duplicated) sample and follow above rules. For example:
> $$
> \mathbf{query}  \Rightarrow  \mathbf{tp}^{1,0} ; \mathbf{fp}^2 ; \mathbf{fp}^3 ; \left[\mathbf{tp}^{4,2} ; \mathbf{tp}^{5,2}\right] ; \mathbf{fp}^6 ;  \mathbf{tp}^{7,3} ; \mathbf{fp}^8
> $$
> The above rank list has two duplicated true positives ($d\left(\mathbf{q}, \mathbf{tp}^{4,2}\right) = d\left(\mathbf{q}, \mathbf{tp}^{5,2}\right)$), then, if a swap happens between them and $\mathbf{fp}^6$, it will become:
> $$
> \mathbf{query}  \Rightarrow  \mathbf{tp}^{1,0} ; \mathbf{fp}^2 ; \mathbf{fp}^3  \mathbf{fp}^4  ; \left[\mathbf{tp}^{5,3} ; \mathbf{tp}^{6,3}\right] ; \mathbf{tp}^{7,3} ; \mathbf{fp}^8
> $$
> where $i, m$ of two duplicated true positives are both increased by $1$. Obviously, the lower bound is still tight.
> * **Case 2**: Mixed positives.\
> It is tricky when true positives and false positives have the same distance with query (we call them mixed positives). The sorting algorithm to produce rank list also has impact to determine ranks of these mixed positives. To determine whether the lower bound is tight in this case is hard, but our lower bound is still valid since `line 38 ~ 43` in `Supp. Sec. B` still make sense.
> * **Case 3**: Rank list with gaps.\
> If there are gaps in between two distances, *e.g.*, $d\left(\mathbf{q}, \mathbf{fp}^i\right)$ is very small but $d\left(\mathbf{q}, \mathbf{tp}^{i+1}\right)$ is very large (this usually happens on outliers), then AP will be far from the lower bound. Please refer to `Q3, edge case 2` for details.
>
> In conclusion, under assumptions of `1, 2, 3`, our lower bound is tight. Meanwhile, our lower bound also covers common cases in above discussion and makes a strong connection to AP.
>
> ---
>
> \
> Please refer to the second part for `Q2 ~ Q5`.

---

> ### Author Response · Authors · 2022-08-08
> **We are happy to address any further concerns**
>
> Dear Reviewer **jH1U**,
>
> We sincerely appreciate the reviewer's effort and constructive comments. We have provided technical details and analysis to clarify your questions. If you have any further concerns, please feel free to let us know and we are more than happy to answer them.
>
>
> \
> \
> Best,
>
> Paper 44 Authors

---

### Official Review · Reviewer_N6zv · 2022-07-12

**Rating:** 3
**Confidence:** 4
**Soundness:** 2 fair
**Presentation:** 1 poor
**Contribution:** 2 fair

**Summary:**

The paper on hand addresses the problem if hashing indicating that interclass distinctiveness and intraclass compactness determine the lower bound of hand code performance. Based on these assumptions a model is proposed, showing beneficial properties for different applications.

**Questions:**

I would argue with benefits in terms of speed but not on "carbon neutrality"

As seen above from the weaknesses, there are several points that would need to be addressed in a revised version to increase the clarity and the reading flow. In particular, the technical contribution needs to discussed in more clear way.

Overall, there are too many flaws for accepting the paper for NeurIPS.

**Limitations:**

The paper on hand is a theoretical contribution, thus there are no limitations in this context.



**Strengths And Weaknesses:**

Strengths:

In general, using hashing could be beneficial for many applications.

The presented results indicate that the approach is beneficial for at least three different applications.


Weaknesses:

There is no clear description of related work. Even though there is a large number of of references, these are quite general and not directly related to the problem on hand.

To fully understand the problem, the preliminary section need to be extended. Neither the overall problem, nor the required technical details are given on an adequate level.

In general, the reading flow is hampered by inadequate structure and mathematical writing. The (wrong) overuse of definitions, propositions, and corollary prevents a fluently reading the paper.

In addition, the mathematical writing needs to be improved. This includes the correct embedding of of equations into the text, missing or insufficiently defined mathematical terms, and a proper argumentation flow.

In general, the paper would benefit from a careful proofreading. There are countless typographical and grammatical errors, hampering the reading flow.

The structure of the paper needs to improved. In particular, the introduction, the conclusion, and the discussion need to be structured differently. The argumentation is not straight forward and somehow redundant.

The caption of Fig. 4 needs to be extended. Even though described in the text, from the figure it is not clear what is shown.

The meaning of Fig. 5 is not fully clear. In this case a more thorough discussion in the text is necessarily required.The

---

> ### Author Response · Authors · 2022-08-02
> **Response to Reviewer N6zv**
>
>
> Thanks for your critical review. Considering your suggestions in review, we hope following reponse could help you for understanding our work and demystifying your concerns.
>
> Actually, reviewer ***jH1U*** thinks our paper **already has a nice presentation and organization**. Both the other two reviewers mark our work with ***good*** soundness, presentation and contribution.
>
> ### ***Q1: Paper structure needs to be improved.***
> **A:** According to your suggestions, we have polished our paper. Please check the uploaded rebuttal revision for changes. Specifically, we add the **Related Works** section to include and describe current advances in learning-to-hash with proper references. We also update descriptions in **Preliminaries** section and successive sections to inrease readability and make the problem we focus on easier to be understood. We have also revised Introduction, Conclusion sections and fixed typos and grammatical errors in paper.
>
> ---
>
> ### ***Q2: Mathematical writing needs to be improved.***
> **A:** We provide a thorough derivation on the proposition of lower bound. All definitions, corollary and propositions are required to formulate the lower-bound in general. We think we have clearly stated the argumentation flow. Specifically, we first start with an arbitrary rank list and study how AP is influenced by true positives and false positives by introducing *mis-rank* (`line 101 ~ 110`). We then give lower bound of AP by extending mis-rank to $\min{d\left(\mathbf{q}, \mathbf{tp}\right)}$ and $\max{d\left(\mathbf{q}, \mathbf{fp}\right)}$ (`line 111 ~ 114`). Finally, we generalize above two distances to inter-class distinctiveness and intra-class compactness and propose the final lower bound (`line 115 ~ 125`). Detailed proof is placed in `Supp. Secs. A and B`. If you have any confusion with above derivation, please point out for us to improve it.
>
> ---
>
> ### ***Q3: Minor issues.***
> **A:** Benefits in terms of "carbon neutrality" are based on the consensus of high efficiency and low cost in downstream hash-based tasks such as fast retrieval `[1]`. To obtain nearest neighbor of a query code as retrieved result, similarities between hash codes are obtained by performing `XOR` operation, which is highly optimized with extremely low energy `[2]`, and therefore reduces power consumption and benefits for carbon neutrality. Other issues such as figure descriptions are revised, please check them in the uploaded revision.
>
> ---
>
>
> ### **References**
> [1] X. Luo, H. Wang, D. Wu, C. Chen, M. Deng, J. Huang, and X.-S. Hua. **A survey on deep hashing methods**. *ACM Trans. Knowl. Discov. Data*, 2022.\
> [2] H. Naseri and S. Timarchi, **Low-power and fast full adder by exploring new XOR and XNOR gates**. *IEEE Trans. Very Large Scale Integr. Syst.* 26(8): 1481-1493, 2018.
>
> \
> \
> Best,
>
> Paper 44 Authors

---

> ### Author Response · Authors · 2022-08-08
> **We are happy to address any further concerns**
>
> Dear Reviewer **N6zv**,
>
> We sincerely appreciate the reviewer's effort and constructive comments. We have polished the paper to clarify your questions. If you have any further concerns, please feel free to let us know and we are more than happy to answer them.
>
>
> \
> \
> Best,
>
> Paper 44 Authors

---

### Official Review · Reviewer_HCrQ · 2022-07-15

**Rating:** 7
**Confidence:** 4
**Soundness:** 3 good
**Presentation:** 3 good
**Contribution:** 3 good

**Summary:**

This paper first proves that inter-class distinctiveness and intra-class compactness among hash codes determine the lower bound of hash codes’ performance. And it shows that promoting these two characteristics could lift the bound and improve hash learning. Then it proposes a surrogate model to fully exploit such objective by estimating posterior of hash codes. Extensive experiments reveal effectiveness of the proposed method.

**Questions:**

Is the performance improvement still significant when we increase the size of training set?

**Limitations:**

The authors have adequately addressed the limitations and potential negative societal impact of their work.

**Strengths And Weaknesses:**

Strengths:
1. The studied problem is interesting and important because a theoretical analysis on criteria of learning good hash codes remains largely unexplored.
2. The proposed method seems to be reasonable and effective.
3. Experiment seems to be extensive.

Weaknesses:
1. There exist some typos and grammatical errors in the paper.
2. The training sets on all datasets are relatively small.

---

> ### Author Response · Authors · 2022-08-02
> **Response to Reviewer HCrQ**
>
>
> Thanks for your kind review and recognize the effectiveness of our work. We would like to provide following response to your major concerns.
>
> ### ***Q1: Is the performance improvement still significant when we increase the size of training set?***
> **A:** We follow public benchmark to evaluate performance of hash models (`Sec. 6.1.2` in main paper). Dataset settings are the same with all methods evaluated in paper. According to your valuable suggestion, we conduct a new experiments on the whole ImageNet dataset (`ILSVRC 2012`) with $1,000$ classes to evaluate performance with large training set ($1.2M$ images, $\sim 10\times$ larger than training sets adopted in main paper). For retrieval, we adopt ImageNet val set which has $50,000$ images and $50$ for each class. We randomly pick $5$ images of each class as queries ($5,000$ in total) and remaining is adopted to formulate base split ($45,000$ in total). We train networks for $20$ epoch and only update the last hash layers since backbone is already pre-trained on ImageNet. Other settings are same with main paper.
>
> Due to time limitation, we report `CSQ` and `CSQ_D` results at present. Other results will be added in camera-ready version.
>
> | Methods |`16 bits`|`32 bits`|`64 bits`|
> |:-------:|:--------|:--------|:--------|
> |   CSQ   | $64.6$                         | $65.0$                         | $65.7$                         |
> |  CSQ_D  | $\mathbf{65.7}_{\uparrow 1.1}$ | $\mathbf{66.2}_{\uparrow 1.2}$ | $\mathbf{66.4}_{\uparrow 0.7}$ |
>
> We could see for `CSQ`, our performance improvement is still valid under the big training set setting.
>
> ---
>
> Other minor issues such as typos and grammatical errors are revised. Please refer to the rebuttal revision of paper. Thanks for your kind remind.
>
> \
> \
> Best,
>
> Paper 44 Authors

---

> > ### Author Response · Authors · 2022-08-05
> > **Update**
> >
> > Now, we update results with `HashNet` as baseline under above setting:
> >
> >
> > | Methods |`16 bits`|`32 bits`|`64 bits`|
> > |:-------:|:--------|:--------|:--------|
> > |   HashNet   | $36.9$                         | $41.2$                         | $44.7$                         |
> > |  HashNet_D  | $\mathbf{55.0}_{\uparrow 18.1}$ | $\mathbf{56.1}_{\uparrow 14.9}$ | $\mathbf{61.8}_{\uparrow 17.1}$ |
> >
> > which also confirms effectiveness of our method.
> >
> > We are evaluating remaining methods. If further results are avaliable, we will update here as soon as possible.
> >
> > \
> > \
> > Best,
> >
> > Paper 44 Authors

---

> > > ### Author Response · Authors · 2022-08-09
> > > **Results of all methods**
> > >
> > > We have conducted all experiments based on above settings. Here, we provide the full table:
> > >
> > > | Methods     | `16 bits`              | `32 bits` | `64 bits` |
> > > |-------------|------------------------|-----------|-----------|
> > > | HashNet     | $36.9$                 | $41.2$    | $44.7$    |
> > > | HashNet_D   | $55.0_{\uparrow 18.1}$ |  $56.1_{\uparrow 14.9}$         |  $61.8_{\uparrow 17.1}$         |
> > > | DBDH        |   $37.4$                     |   $41.1$        |   $49.1$        |
> > > | DBDH_D      |   $57.2_{\uparrow 19.8}$                     |  $57.9_{\uparrow 16.8}$         |   $60.4_{\uparrow 11.3}$        |
> > > | DSDH        |     $39.3$                   |  $48.4$         |    $54.2$       |
> > > | DSDH_D      |   $56.0_{\uparrow 16.7}$                     |  $59.1_{\uparrow 10.7}$         |   $62.0_{\uparrow 7.8}$        |
> > > | DCH         |     $60.2$                   |   $62.3$        |   $64.1$        |
> > > | DCH_D       |   $61.9_{\uparrow 1.7}$                     |     $62.9_{\uparrow 0.6}$      |    $64.4_{\uparrow 0.3}$       |
> > > | GreedHash   |    $62.7$                    |    $63.1$       |   $64.9$        |
> > > | GreedHash_D |    $63.1_{\uparrow 0.4}$                    |   $64.0_{\uparrow 0.9}$        |   $65.9_{\uparrow 1.0}$        |
> > > | CSQ         |        $64.6$                |     $65.0$      |    $65.7$       |
> > > | CSQ_D       |       $65.7_{\uparrow 1.1}$                 |     $66.2_{\uparrow 1.2}$      |  $66.4_{\uparrow 0.7}$         |
> > >
> > > You could also check it in the updated `Supp. Sec. F`.
> > >
> > > \
> > > \
> > > Best,
> > >
> > > Paper 44 Authors

---

> > > > ### Comment · Reviewer_HCrQ · 2022-08-09
> > > > **my concern on experiment has been addressed**
> > > >
> > > > Thanks for the response. My concern on experiment has been addressed. The writing need to be further polished.

---

> > > > > ### Author Response · Authors · 2022-08-10
> > > > > **Thanks for your suggestion**
> > > > >
> > > > > We are happy to make this paper better. Actually, we have already polished several sections and improved readability in the rebuttal revision (please refer to `General Response`). Considering on your suggestion, we will go on a further proofreading to check if there needs improvements of the paper.
> > > > >
> > > > > \
> > > > > \
> > > > > Best,
> > > > >
> > > > > Paper 44 Authors

---

> ### Author Response · Authors · 2022-08-08
> **We are happy to address any further concerns**
>
> Dear Reviewer **HCrQ**,
>
> We sincerely appreciate the reviewer's effort and constructive comments. We have polished the paper and provided further experiments to clarify your questions. If you have any further concerns, please feel free to let us know and we are more than happy to answer them.
>
>
> \
> \
> Best,
>
> Paper 44 Authors

---

### Author Response · Authors · 2022-08-02
**General Response**

We are appreciated to reviewers **HCrQ**, **N6zv** and **jH1U** for your kind reviews, and recognizing our work benefits for learning-to-hash methods.

According to reviews, **highlights of this paper include**:
* The proposed lower bound and its theoretical analysis are interesting and important. (**HCrQ**, **jH1U**)
* The proposed method is reasonable and effective, and experiment is extensive. (**HCrQ**, **jH1U**)
* The paper has a nice presentation and organization. (**jH1U**)

Both of reviewers **HCrQ** and **jH1U** mark our work as **good** soundness, presentation and contribution. They suggest acceptance.

**Major weakness of this paper is writing issues**, including:
* Related works and preliminaries need to be extended to fully understand the problem.
* The readability of paper needs to be improved.

Based on above summary, the rebuttal version of paper has been uploaded. Specifically,

* We add the ***Related Works*** section to describe current advances in learning-to-hash.
* We polish the ***Preliminaries*** section to make the problem easier to be understood.
* Section 4 is simplified to increase readability.

We have also added analysis from the response to Reviewer **jH1U**, experiments from the response to Reviewer **HCrQ** in supplementary materials.

\
We hope above changes could make the paper better.

\
\
Best,

Paper 44 Authors

---

### Meta-Review · Area_Chair_mRSV · 2022-09-06

**Recommendation:** Accept
**Confidence:** Certain

**Metareview:**

The paper proposed an interesting lower bound in the learning to hash scenarios and builds on that to show a good algorithm that outperforms several learning to hash methods. There were concerns about the size and scale of experiments which was sufficiently addressed in the rebuttal. The reviewers were not in consensus primarily because of the writing. We think that the writing concerns are fixable and the authors will improve the draft using the reviewers comment for the final version.

**Award:**

No

---

### Decision · Program_Chairs · 2022-09-14

Accept